# Pathologic polyglutamine aggregation begins with a self-poisoning polymer crystal

Tej Kandola[1,2†], Shriram Venkatesan[1†], Jiahui Zhang[3], Brooklyn T Lerbakken[1], Alex Von Schulze[1], Jillian F Blanck[1], Jianzheng Wu[1,4], Jay R Unruh[1], Paula Berry[1], Jeffrey J Lange[1], Andrew C Box[1], Malcolm Cook[1], Celeste Sagui[3], Randal Halfmann[1]*

[1]Stowers Institute for Medical Research, Kansas City, United States; [2]The Open University, Milton Keynes, United Kingdom; [3]Department of Physics, North Carolina State University, Raleigh, United States; [4]Department of Biochemistry and Molecular Biology, University of Kansas Medical Center, Kansas City, United States

*For correspondence:
rhn@stowers.org

†These authors contributed equally to this work

Competing interest: The authors declare that no competing interests exist.

## Abstract

A long-standing goal of amyloid research has been to characterize the structural basis of the rate-determining nucleating event. However, the ephemeral nature of nucleation has made this goal unachievable with existing biochemistry, structural biology, and computational approaches. Here, we addressed that limitation for polyglutamine (polyQ), a polypeptide sequence that causes Huntington's and other amyloid-associated neurodegenerative diseases when its length exceeds a characteristic threshold. To identify essential features of the polyQ amyloid nucleus, we used a direct intracellular reporter of self-association to quantify frequencies of amyloid appearance as a function of concentration, conformational templates, and rational polyQ sequence permutations. We found that nucleation of pathologically expanded polyQ involves segments of three glutamine (Q) residues at every other position. We demonstrate using molecular simulations that this pattern encodes a four-stranded steric zipper with interdigitated Q side chains. Once formed, the zipper poisoned its own growth by engaging naive polypeptides on orthogonal faces, in a fashion characteristic of polymer crystals with intramolecular nuclei. We further show that self-poisoning can be exploited to block amyloid formation, by genetically oligomerizing polyQ prior to nucleation. By uncovering the physical nature of the rate-limiting event for polyQ aggregation in cells, our findings elucidate the molecular etiology of polyQ diseases.

## eLife assessment

The authors investigate the mechanism of amyloid nucleation in a cellular system using novel ratio-metric measurements, providing **fundamental** insight into the role of polyglutamine length and the sequence features of glutamine-rich regions in amyloid formation. The problem addressed by this study is very significant and the ability to assess nucleation in cells is of considerable value. The data, as presented and analyzed, are mostly **convincing**.

## Introduction

Amyloids are highly ordered protein aggregates with self-templating activity. This activity drives the progression of multiple incurable diseases of aging, such as Alzheimer's (*Chiti and Dobson, 2017*; *Huang et al., 2019*). Understanding how amyloids start, or nucleate, is therefore fundamental to

**eLife digest** Diseases that typically occur later in life, such as Alzheimer's, are often caused by specific proteins clumping together into structures known as amyloids. Once the process starts, amyloids will continue to form, leading to worse symptoms that cannot be cured. The best way to treat these diseases is therefore to stop amyloids from arising in the first place.

Amyloids initially develop by proteins coming together to create an unstable structure referred to as the nucleus. The instability of the nucleus means it cannot be observed directly, making it hard to study this nucleation process. To overcome this, Kandola, Venkatesan et al. investigated the simplest protein known to form an amyloid – polyglutamine, which is made up of a chain of repeating building blocks known as amino acids.

Polyglutamine forms only one type of amyloid which is associated with nine neurodegenerative diseases, including Huntington's disease. However, it only does this when its chain of amino acids exceeds a certain length, suggesting that a specific structure may be required for nucleation to begin.

Kandola, Venkatesan et al. made alternative versions of the polyglutamine protein which each contained slightly different sequences of amino acids that will alter the way the protein folds. They then tested how well these different variants could form amyloids in yeast cells. This revealed that in order to join together into a nucleus, polyglutamine needs to be able to fold into a zipper shape made up of four interlocking strands. The length of the protein required to form this shape is also the same length that causes the amyloid associated with neurodegenerative diseases.

Kandola, Venkatesan et al. also found that polyglutamine tends to bind to nuclei that have already formed in a way that hinders their growth. This 'self-poisoning' affect could potentially be exploited as a way to pre-emptively stop amyloids from initially arising.

These findings have uncovered a potential therapeutic strategy for blocking amyloid formation that could eventually benefit people with or at risk of developing neurodegenerative diseases linked to polyglutamine. Additionally, this approach provides a blueprint for understanding how other proteins undergo amyloid nucleation, including those responsible for Alzheimer's, Parkinson's, and other diseases.

preventing these diseases. However, despite decades of intense research, we still lack a clear picture of even the gross anatomy of an amyloid nucleus.

Polyglutamine (polyQ) is an amyloid-forming sequence common to eukaryotic proteomes (*Mier et al., 2020*). In humans, it is responsible for nine invariably fatal neurodegenerative diseases, the most prevalent of which is Huntington's Disease. Cells exhibiting polyQ pathology, whether in Huntington's disease patients, tissue culture, organotypic brain slices, or animal models, die independently and stochastically with a constant frequency (*Clarke et al., 2000*; *Linsley et al., 2019*). Two observations suggest that this kinetic feature emerges directly from the amyloid nucleation barrier. First, polyQ aggregation itself occurs stochastically in cells (*Colby et al., 2006*; *Kakkar et al., 2016*; *Sinnige et al., 2021*). Second, polyQ disease onset and progression are determined almost entirely by an *intramolecular* change, specifically, a genetically encoded expansion in the number of sequential glutamines beyond a protein-specific threshold of approximately 36 residues (*Lieberman et al., 2019*). Unlike other amyloid diseases (*Book et al., 2018*; *Kim et al., 2020*; *Selkoe and Hardy, 2016*), polyQ disease severity generally does not worsen with gene dosage (*Cubo et al., 2019*; *Lee et al., 2019*; *Wexler et al., 1987*), implying that the rate-determining step in neuronal death occurs in a fixed minor fraction of the polyglutamine molecules. It is unclear how, and why, those molecules differ from the bulk. Hence, more so than for other proteopathies, therapeutic progress against polyQ diseases awaits detailed knowledge of the very earliest steps of amyloid formation.

Amyloid nuclei cannot be observed directly by any existing experimental approach. This is because unlike mature amyloid fibrils, which are stable and amenable to structural biology, nuclei are unstable by definition. Their structures do not necessarily propagate into or correspond with the structures of mature amyloids that arise from them (*Auer et al., 2008*; *Buell, 2017*; *Erdemir et al., 2009*; *Hsieh et al., 2017*; *Levin et al., 2014*; *Liang et al., 2018*; *Li et al., 2010*; *Phan and Schmit, 2020*; *Sil et al., 2018*; *Yamaguchi et al., 2005*; *Zanjani et al., 2020*). Moreover, amyloid nucleation by a full length polypeptide occurs far too infrequently, and involves far too many degrees of freedom, to simulate

with high resolution from a naive state (*Barrera et al., 2021*; *Kar et al., 2011*; *Strodel, 2021*). Unlike phase separation, which primarily concerns a loss of *intermolecular* entropy, amyloid formation additionally involves a major loss of *intramolecular* entropy. That is, nucleation selects for a specific combination of backbone and side chain torsion angles (*Khan et al., 2018*; *Vitalis and Pappu, 2011*; *Zhang and Schmit, 2016c*). As a consequence, amyloid-forming proteins can accumulate to supersaturating concentrations while remaining soluble, thereby storing potential energy that will subsequently drive their aggregation following a stochastic nucleating event (*Buell, 2017*; *Khan et al., 2018*).

Due to the improbability that the requisite increases in both density and conformational ordering will occur spontaneously at the same time (i.e. *homogeneously*), amyloid nucleation tends to occur *heterogeneously*. In other words, amyloids tend to emerge via a progression of relatively less ordered but more probable, metastable intermediates of varying stoichiometry and conformation (*Auer et al., 2008*; *Buell, 2017*; *Hsieh et al., 2017*; *Levin et al., 2014*; *Liang et al., 2018*; *Li et al., 2010*; *Serio et al., 2000*; *Sil et al., 2018*; *Vekilov, 2012*; *Vitalis and Pappu, 2011*; *Yamaguchi et al., 2005*; *Zanjani et al., 2020*). A major body of current opinion holds that such heterogeneities tend to involve liquid-liquid phase separation, whereby the protein first forms disordered multimers that then facilitate productive conformational fluctuations (*Borcherds et al., 2021*; *Camino et al., 2021*; *Crick et al., 2013*; *Crick et al., 2006*; *Dignon et al., 2020*; *Fisher et al., 2021*; *Halfmann et al., 2011*; *Peskett et al., 2018*; *Posey et al., 2018*; *Vitalis and Pappu, 2011*; *Yang and Yang, 2020*). Heterogeneities divide the nucleation barrier into smaller, more probable steps. Therefore, the occurrence of heterogeneities implies that the nature of the actual nucleus for a given amyloid can depend not only on the protein's sequence but also its *concentration* and cellular factors that influence its *conformation* (*Bradley et al., 2002*; *Buell, 2017*; *Collinge and Clarke, 2007*; *Phan and Schmit, 2022*; *Sanders et al., 2014*; *Törnquist et al., 2018*), as illustrated in *Figure 1A*.

Heterogeneities may be responsible for amyloid-associated proteotoxicity. Partially ordered species accumulate during the early stages of amyloid aggregation by all well-characterized pathological amyloids, but generally do not occur (or less so) during the formation of functional amyloids (*Otzen and Riek, 2019*). In the case of pathologically expanded polyQ, or the huntingtin protein containing pathologically expanded polyQ, amyloid-associated oligomers have been observed in vitro, in cultured cells, and in the brains of patients (*Auer et al., 2008*; *Hsieh et al., 2017*; *Legleiter et al., 2010*; *Levin et al., 2014*; *Liang et al., 2018*; *Li et al., 2010*; *Olshina et al., 2010*; *Sathasivam et al., 2010*; *Sil et al., 2018*; *Takahashi et al., 2008*; *Vitalis and Pappu, 2011*; *Yamaguchi et al., 2005*; *Zanjani et al., 2020*), and are likely culprits of proteotoxicity (*Kim et al., 2016*; *Leitman et al., 2013*; *Lu and Palacino, 2013*; *Matlahov and van der Wel, 2019*; *Miller et al., 2011*; *Takahashi et al., 2008*; *Wetzel, 2020*). In contrast, mature amyloid fibers are increasingly viewed as relatively benign or even protective (*Arrasate et al., 2004*; *Kim et al., 2016*; *Leitman et al., 2013*; *Lu and Palacino, 2013*; *Mario Isas et al., 2021*; Matlahov and *Matlahov and van der Wel, 2019*; *Takahashi et al., 2008*; *Wetzel, 2020*). The fact that polyQ disease kinetics is governed by amyloid nucleation, but the amyloids themselves may not be responsible, presents a paradox. A structural model of the polyQ amyloid nucleus can be expected to resolve this paradox by illuminating a path for the propagation of conformational order into distinct multimeric species that either cause or mitigate toxicity.

Structural features of amyloid nucleation may be deduced by studying the effects on amyloid kinetics of rational mutations made to the polypeptide (*Thakur and Wetzel, 2002*). The occurrence of density heterogeneities confounds this approach, however, as they blunt the dependence of amyloid kinetics on concentration that could otherwise reveal nucleus stoichiometry (*Michaels et al., 2023*; *Vitalis and Pappu, 2011*). Resolving this problem would require an assay that can detect nucleation events independently of amyloid growth kinetics, and which can scale to accommodate the large numbers of mutations necessary to identify sequence-structure relationships. Classic assays of in vitro amyloid assembly kinetics are poorly suited to this task, both because of their limited throughput, and because experimentally tractable reaction volumes are typically too large to observe discrete aggregation events (*Michaels et al., 2017*).

We recently developed an assay to circumvent these limitations (*Khan et al., 2018*; *Posey et al., 2021*; *Venkatesan et al., 2019*). Distributed Amphifluoric FRET (DAmFRET) uses a monomeric photoconvertible fusion tag and high throughput flow cytometry to treat living cells as femtoliter-volume test tubes, thereby providing the large numbers of independent reaction vessels of exceptionally small volume that are required to discriminate independent nucleation events under physiological

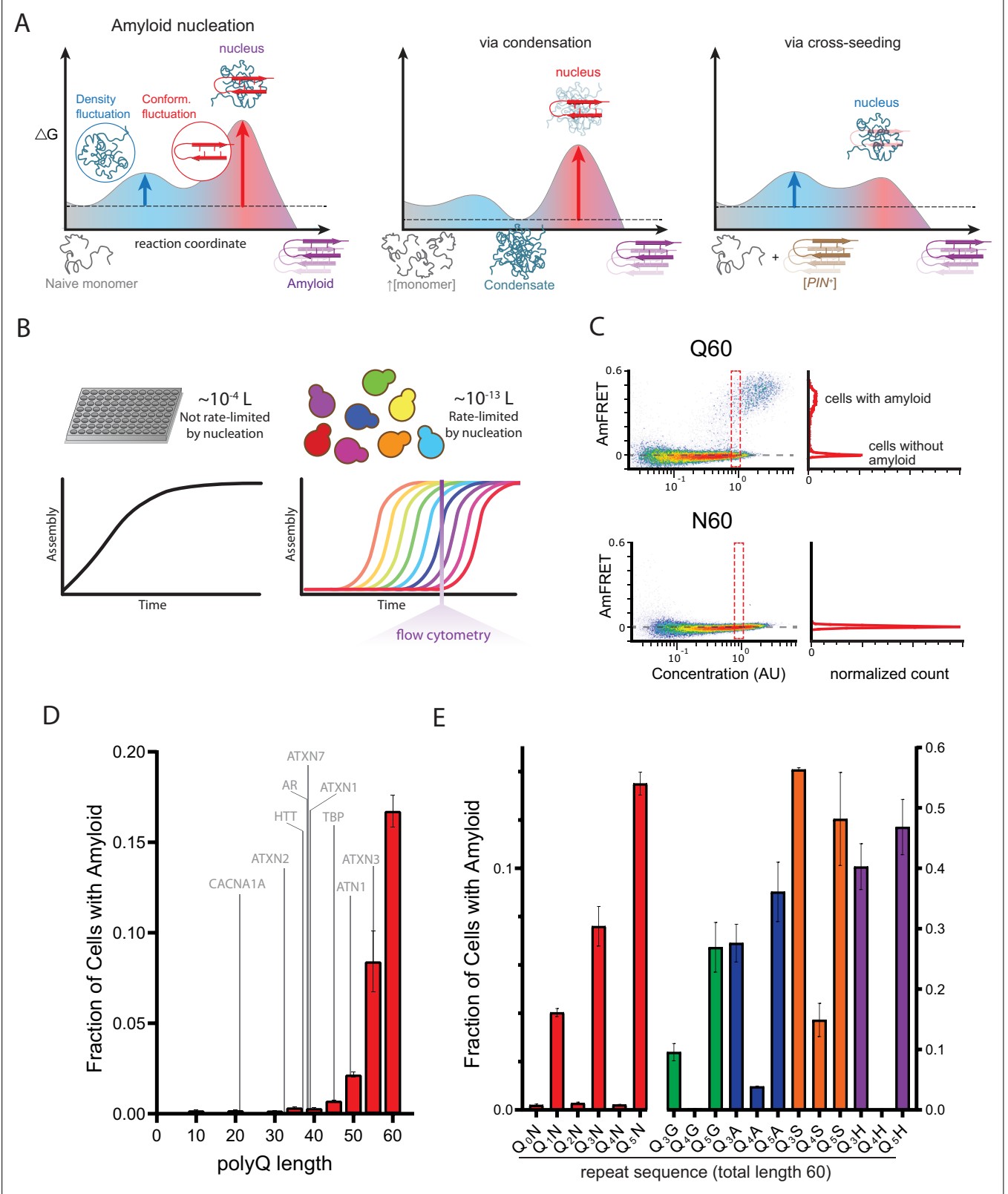

**Figure 1.** A hidden pattern of glutamines governs amyloid nucleation. (**A**) Reaction coordinate diagrams schematizing the energy barriers governing amyloid nucleation. Because amyloid involves a transition from disordered monomer to ordered multimer, the nucleation barrier results from a combination of high energy fluctuations in both density (blue) and conformation (red). The present study manipulates either component of the nucleation barrier by changing the protein's concentration (middle panel, illustrating a higher concentration surpassing a phase boundary for

*Figure 1 continued on next page*

Figure 1 continued

condensation) or the presence/absence of a conformational heterogeneity (right panel, with the amyloid template, [*PIN*⁺]). Nucleation then depends on the remaining fluctuation, as illustrated by vertical arrows. Cartoon images illustrate the relevant reactant species (naive monomers, condensate, [*PIN*⁺]), products (amyloid), and transition states (nuclei). Note that while the relative heights of the blue and red barriers are arbitrary, we illustrate the latter as higher in keeping with the findings of *Khan et al., 2018* for prion-like amyloid nucleation. (**B**) Cellular volumes quantize amyloid nucleation. Amyloid nuclei occur at such low concentrations that fewer than one exists in the femtoliter volumes of cells. This causes amyloid formation to be rate-limited by stochastic nucleation in individual yeast cells (right) but not in the microliter volumes of conventional in vitro kinetic assays (left). Taking a population-level snapshot of the extent of protein self-assembly as a function of concentration in each cell reveals heterogeneity attributable to the nucleation barrier. (**C**) DAmFRET plots, showing the extent of de novo self-assembly (AmFRET) as a function of protein concentration for polyglutamine (**Q60**) or polyasparagine (**N60**) in yeast cells, lacking endogenous amyloid ([*pin*⁻]). Cells expressing Q60 partition into distinct populations that either lack (no AmFRET) or contain (high AmFRET) amyloid. The bimodal distribution persists even among cells with the same concentration of protein, indicating that amyloid formation is rate-limited by nucleation. The nucleation barrier for N60 is so large that spontaneous amyloid formation occurs at undetectable frequencies. Insets show histograms of AmFRET values. AU, arbitrary units. (**D**) Bar plot of the fraction of [*pin*⁻] cells in the AmFRET-positive population for the indicated length variants of polyQ, along with the pathologic length thresholds for polyQ tracts in the indicated proteins. Shown are means +/- SEM of biological triplicates. CACNA1A, Cav2.1; ATXN2, Ataxin-2; HTT, Huntingtin; AR, Androgen Receptor; ATXN7, Ataxin-7; ATXN1, Ataxin-1; TBP, TATA-binding protein; ATN1, Atrophin-1; ATXN3, Ataxin-3. (**E**) Bar plot of the fraction of [*pin*⁻] cells in the AmFRET-positive population for the indicated sequences, showing that amyloid is inhibited when Q tracts are interrupted by a non-Q side chain at odd-numbered intervals. Shown are means +/- SEM of biological triplicates.

The online version of this article includes the following figure supplement(s) for figure 1:

**Figure supplement 1.** DAmFRET plots of polyQ sequence variants.

**Figure supplement 2.** DAmFRET plots of N-rich and QnX sequence variants.

conditions (*Figure 1B*). Compared to conventional reaction volumes for protein self-assembly assays, the budding yeast cells employed in DAmFRET increase the dependence of amyloid formation on nucleation by nine orders of magnitude (the difference in volumes), allowing amyloid to form in the same nucleation-limited fashion as it does in afflicted neurons (*Colby et al., 2006*; *Wetzel, 2006*). DAmFRET employs a high-variance expression system to induce a hundred-fold range of concentrations of the protein of interest. The large intramolecular fluctuation required for amyloid nucleation manifests as a discontinuity in the relationship of ratiometric FRET ('AmFRET') to concentration, and in this way allows DAmFRET to distinguish amyloid formation from finite oligomerization and liquid-liquid phase separation. AmFRET values report the FRET intensity per molecule of protein; hence they plateau with concentration as the fraction of molecules in a given self-assembled state approaches unity. Unlike in classical in vitro kinetic experiments, where the protein concentration does not vary with time, in typical DAmFRET experiments, the concentration of protein increases monotonically with time in each cell (*Li et al., 2000*; *Posey et al., 2021*) over the hours-long induction prior to analysis. Nucleation occurs semi-stochastically during this time, such that the measured fraction of cells with amyloid at a given concentration reflects the cumulative probability of nucleation while transiting all lower concentrations. One can therefore qualitatively infer how nucleation depends on concentration from single DAmFRET plots. In short, by taking a snapshot of the protein's concentration and extent of self-association in each cell, at a point in time appropriate for the kinetics of amyloid formation (16 hr), DAmFRET probes the existence and magnitude of critical density fluctuations that may govern nucleation.

The yeast system additionally allows for orthogonal experimental control over the critical conformational fluctuation. This is because yeast cells normally contain exactly one cytosolic amyloid species—a prion state formed by the low abundance Q-rich endogenous protein, Rnq1 (*Kryndushkin et al., 2013*; *Nizhnikov et al., 2014*). The prion state can be gained or eliminated experimentally to produce cells whose sole difference is whether the Rnq1 protein does ([*PIN*⁺]) or does not ([*pin*⁻]) populate an amyloid conformation (*Derkatch et al., 2001*). [*PIN*⁺] serves as a partial template for amyloid formation by compositionally similar proteins including polyQ (*Alexandrov et al., 2008*; *Duennwald et al., 2006a*; *Meriin et al., 2002*; *Serpionov et al., 2015*). Known as cross-seeding, this phenomenon is analogous to, but much less efficient than, amyloid elongation by molecularly identical species (*Keefer et al., 2017*; *Khan et al., 2018*; *Serio, 2018*). By evaluating nucleation frequencies as a function of concentration in both [*pin*⁻] and [*PIN*⁺] cells, we can uncouple the two components of the nucleation barrier and thereby relate specific sequence features to the nucleating conformation. Sequence changes that are specific to the sequence-encoded nucleus will more strongly influence nucleation in [*pin*⁻] cells than in [*PIN*⁺] cells.

PolyQ is an ideal polypeptide with which to deduce the physical nature of a pathologic amyloid nucleus. It has zero complexity, which simplifies the design and interpretation of sequence variants. Unlike other pathogenic protein amyloids, which occur as different structural polymorphs each plausibly with their own structural nucleus, polyQ amyloids have an invariant core structure under different assembly conditions and in the context of different flanking domains (*Boatz et al., 2020*; *Galaz-Montoya et al., 2021*; *Lin et al., 2017*; *Schneider et al., 2011*). This core contains antiparallel β-sheets (*Buchanan et al., 2014*; Matlahov and *Matlahov and van der Wel, 2019*; *Schneider et al., 2011*), while most other amyloids, including all other Q-rich amyloids, have a parallel β-sheet core (*Eisenberg and Jucker, 2012*; *Margittai and Langen, 2008*). This suggests that amyloid nucleation and pathogenesis of polyQ may follow from an ability to spontaneously acquire a *specific* nucleating conformation. Any variant that decelerates nucleation can therefore be interpreted with respect to its effect on *that* conformation.

The experiments and results reported herein reveal that the polyQ amyloid nucleus is a steric zipper encoded by a pattern of approximately twelve Q residues in a single polypeptide molecule. We find that the clinical length threshold for polyQ disease—approximately 36 residues—is the minimum length that encompasses the pattern. Consistent with a monomeric nucleus, amyloid formation occurred less frequently at high concentrations or when polyQ was expressed in oligomeric form. We further found that the nucleus promotes its own kinetic arrest by templating competing dimensions of Q zipper ordering—both along the amyloid axis and orthogonally to it. This leads to the accumulation of partially ordered aggregates.

## Results

### A hidden pattern of glutamines governs amyloid nucleation

To validate the use of DAmFRET for pathologic polyQ, we first surveyed the length-dependence for amyloid formation by polyQ tracts expressed as fusions to mEos3.1 in nondividing [*pin*⁻] yeast cells. We genetically compromised protein degradation in these cells to prevent differential turnover of potential polyQ heterogeneities, which could otherwise obscure the relationship of aggregation to concentration. Our data recapitulated the pathologic threshold—Q lengths 35 and shorter lacked AmFRET, indicating a failure to aggregate or even appreciably oligomerize, while Q lengths 40 and longer did acquire AmFRET in a length and concentration-dependent manner (*Figure 1C–D*; *Figure 1—figure supplement 1A*). The values of AmFRET at high concentrations decreased with polyQ length, as expected because longer molecules have a lower mass fraction of fluorophore. The cells partitioned into two discontinuous populations: one with high AmFRET and the other with none (top and bottom populations in *Figure 1C*). The two populations occurred with overlapping concentration ranges (red dashed box in *Figure 1C*, top). This suggests that a kinetic barrier associated with conformational ordering limits the acquisition of the high-FRET state (*Khan et al., 2018*). The two populations were well-resolved at high concentrations, where relatively few cells populated intermediate values of AmFRET. These features together indicate that aggregation of polyQ is rate-limited by a nucleation barrier associated with a rare conformational fluctuation.

Sixty residues proved to be the optimum length to observe both the pre- and post-nucleated states of polyQ in single experiments, and corresponds clinically to disease onset in early adulthood (*Kuiper et al., 2017*). We therefore focused on this length moving forward. The frequency of polyQ nucleation greatly exceeds that of other Q-rich amyloid-forming proteins of comparable length (*Khan et al., 2018*; *Posey et al., 2021*). To determine if the protein's exceptional amyloid propensity can be attributed to a property of the Q residues themselves, as opposed to its extremely low sequence complexity, we next tested a 60-residue homopolymer of asparagine (polyN), whose side chain is one methylene shorter but otherwise identical to that of glutamine. PolyN is known to form amyloid (*Halfmann et al., 2011*). We found that polyN did populate the high FRET state, but at much lower frequencies than polyQ even at the highest concentrations (*Figure 1C*; *Figure 1—figure supplement 1B*), approximately 200 μM (*Khan et al., 2018*).

The apparently much larger nucleation barrier for polyN—despite its physicochemical similarity to polyQ—led us to consider whether substituting Qs for Ns at specific positions in polyQ might reveal patterns that uniquely encode the structure of the polyQ nucleus. We reasoned that a random screen of Q/N sequence space would be unlikely to yield informative patterns, however, given that there are

more than one trillion combinations of Q and N for even a 40-residue peptide. We therefore devised a systematic approach to rationally sample Q/N sequence space. Specifically, we characterized by DAmFRET a series of related sequences with a single N residue inserted after every $q$ Q residues over a total length of 60, for all values of $q$ from 0 through 9 (designated $Q_1N$, $Q_2N$, etc. with the value of $q$ subscripted; *Supplementary file 1*). The resulting dataset revealed a shockingly strong dependence of amyloid propensity on the exact sequence of Q residues, and specifically the following two determinants.

First, we observed amyloid formation for all values of $q \geq 6$ (*Figure 1—figure supplement 1B*). Second, amyloid formation for values of $q<6$ was limited to odd numbers 1, 3, and 5 (*Figure 1E*). That is, the relationship of amyloid formation to the number of consecutive Qs exhibited an 'odd-even effect' that saturated at six. This simple pattern could be explained by a local requirement for Q at every other position along the sequence (i.e. $i$, $i+2$, $i+4$, etc.). To test this, we inserted a second N residue into the repeats for two such sequences. Indeed, these failed to aggregate (*Figure 1—figure supplement 2A*).

To determine if this 'odd' dependency results from a general preference of amyloid for homogeneity at every other position, irrespective of their amino acid identity, we created an analogous 'polyN' series with a single Q placed after every first, second, third, or fourth N. This series showed no preference for even or odd values (*Figure 1—figure supplement 2A*), confirming that Qs, specifically, are required at every other position. We next asked if the odd-even effect concerns the Q side chains themselves, rather than their interaction with Ns, by replacing the Ns with residues of diverse physicochemistry—either glycine, alanine, serine, or histidine. Again, nucleation was much more frequent for $Q_3X$ and $Q_5X$ than for $Q_4X$ (*Figure 1E*; *Figure 1—figure supplement 2B*), confirming that the odd-even effect resulted from the Q side chains interacting specifically with other Q side chains. The identity of X did, however, influence nucleation frequency particularly for $Q_4X$. Glycine proved just as detrimental as asparagine; histidine slightly less so; and alanine lesser still. Serine was relatively permissive. The relative indifference of $Q_3X$ and $Q_5X$ to the identity of X suggests that these sequences form a pure Q core that excludes every other side chain. In other words, the nucleus is encoded by segments of sequence with Qs at every other position.

To reduce the contribution of the conformational fluctuation to nucleation, we next performed DAmFRET experiments for all sequences in [$PIN^+$] cells. Amyloid formation broadly increased (*Figure 1—figure supplement 1A–B*; *Figure 1—figure supplement 2A–B*), with polyQ now nucleating in most [$PIN^+$] cells even at the lowest concentrations sampled by DAmFRET (~1 μM, *Khan et al., 2018*). Importantly, amyloid formation again occurred only for polyQ lengths exceeding the clinical threshold. Additionally, nucleation in [$PIN^+$] retained the structural constraints of de novo nucleation, as indicated by (a) the relative impacts of different X substitutions, which again increased in the order serine <alanine < histidine <glycine/asparagine; and (b) the persistence of the odd-even effect, with $q$=1, 3, and 5 nucleating more frequently than $q$=4. The latter failed entirely to form amyloid in the cases of asparagine and glycine. All N-predominant sequences and only two Q-predominant sequences ($Q_2N$ and $(Q_3N)_2N$; introduced below) nucleated robustly in [$PIN^+$], even though they failed entirely to do so in [$pin^-$] cells. Their behavior resembles that of typical 'prion-like' sequences (*Khan et al., 2018*; *Posey et al., 2021*), consistent with our deduction that these amyloids necessarily have a different nucleus structure than that of uninterrupted polyQ.

## The pattern encodes a single unique steric zipper

What structure does the pattern encode, and why is it so sensitive to Ns? We sought to uncover a physical basis for the different nucleation propensities between Q and N using state-of-the-art amyloid predictors (*Bryan et al., 2009*; *Burdukiewicz et al., 2017*; *Charoenkwan et al., 2021*; *Conchillo-Solé et al., 2007*; *Emily et al., 2013*; *Fernandez-Escamilla et al., 2004*; *Garbuzynskiy et al., 2010*; *Goldschmidt et al., 2010*; *Hamodrakas et al., 2007*; *Keresztes et al., 2021*; *Kim et al., 2009*; *Maurer-Stroh et al., 2010*; *O'Donnell et al., 2011*; *Prabakaran et al., 2021*; *Thangakani et al., 2014*; *Walsh et al., 2014*). Unfortunately, none of the predictors were able to distinguish $Q_3N$ and $Q_5N$ from $Q_4N$ (*Supplementary file 2*), despite their very dissimilar experimentally determined amyloid propensities. In fact, most predictors failed outright to detect amyloid propensity in this class of sequences. Apparently, polyQ is exceptional among the known amyloid-forming sequences on which these predictors were trained, again hinting at a specific nucleus structure.

Most pathogenic amyloids feature residues that are hydrophobic and/or have a high propensity for β-strands. Glutamine falls into neither of these categories (*Fujiwara et al., 2012*; *Nacar, 2020*; *Simm et al., 2016*). We reasoned that nucleation of polyQ therefore corresponds to the formation of a specific tertiary structure unique to Q residues. Structural investigations of predominantly Q-containing amyloid cores reveal an exquisitely ordered tertiary motif wherein columns of Q side chains from each sheet fully extend and interdigitate with those of the opposing sheet (*Hoop et al., 2016*; *Schneider et al., 2011*; *Sikorski and Atkins, 2005*). The resulting structural element exhibits exceptional shape complementarity (even among amyloids) and is stabilized by the multitude of resulting van der Waals interactions. Additionally, the terminal amides of Q side chains form regular hydrogen bonds to the backbones of the opposing β-sheet (*Esposito et al., 2008*; *Hervas et al., 2020*; *Man et al., 2015*; *Schneider et al., 2011*; *Zhang et al., 2016b*). To minimize confusion while referring to these distinguishing tertiary features of the all-Q amyloid core, relative to the similarly packed cores of virtually all other amyloids, which lack regular side chain-to-backbone H bonds (*Eisenberg and Sawaya, 2017*; *Sawaya et al., 2007*), we will henceforth refer to them as 'Q zippers' (*Figure 2A*). Perhaps the entropic cost of acquiring such an exquisitely ordered structure by the entirely disordered ensemble of naive polyQ (*Chen et al., 2001*; *Moradi et al., 2012*; *Newcombe et al., 2018*; *Vitalis et al., 2007*; *Wang et al., 2006*) underlies the nucleation barrier.

Odd-even effects have long been observed in the crystallization of *n*-alkanes, where the 180° rotation of successive carbons leads to packing differences between odd- and even-lengthed segments (*Dhiman et al., 2022*; *Pérez-Camargo et al., 2021*; *Tao and Bernasek, 2007*). We reasoned that, similarly, the present odd-even effect could relate to the fact that successive side chains alternate by 180° along a β-strand (*Figure 2B–C*). Hence, strands with even values of $q$ will necessarily have non-Q side chains on both sides, preventing either side from participating in a Q zipper. Our experimental results imply that the non-Q residue is most disruptive when it is an N, and much less so when it is an S. To determine if the Q zipper has such selectivity, we carried out fully atomistic molecular dynamics simulations on Q-substituted variants of a minimal Q zipper model (*Zhang et al., 2016b*). Specifically, we aligned four Q7 peptides into a pair of interdigitated two-stranded antiparallel β-sheets with interdigitated Q side chains. We then substituted Q residues in different positions with either N or S and allowed the structures to evolve for 200 ns. In agreement with the experimental data, the Q zipper was highly specific for Q side chains: it rapidly dissolved when any proximal pair of inward pointing Qs were substituted, as would occur for unilaterally discontinuous sequences such as $Q_4X$ (*Figure 2D*; *Figure 2—figure supplement 1A*). In contrast, it remained intact when any number of outward-pointing Q residues were substituted, as would occur for $Q_3X$ and $Q_5X$. Given the sensitivity of this minimal zipper to substitutions, we next constructed a zipper with twice the number of strands, where each strand contained a single inward pointing N or S substitution, and allowed the structures to evolve for 1200 ns. In this case, the S-containing zipper remained intact while the N-containing zipper dissolved (*Figure 2—figure supplement 1B*), consistent with our experimental results.

A close examination of the simulation trajectories revealed how N disrupts Q zippers. When inside a Q zipper, the N side chain is too short to H-bond the opposing backbone. In our simulations, the terminal amide of N competed against the adjacent backbone amides for H-bonding with the terminal amide of an opposing Q side-chain, blocking that side chain from fully extending as required for Q zipper stability (*Figure 2E–F*). The now detached Q side chain collided with adjacent Q side chains, causing them to also detach, ultimately unzipping the zipper (*Figure 2—figure supplement 2A*; *Figure 2—video 1*; *Figure 2—video 2*). We found that the side chain of S, which is slightly shorter and less bulky than that of N, does not intercept H-bonds from the side chains of opposing Q residues (*Figure 2—figure supplement 2B*). As a result, order is maintained so long as multiple S side chains do not occur in close proximity inside the zipper.

We also considered an alternative mechanism whereby N side chains H-bonded to adjacent Q side chains on the same side of the strand (i.e. $i+2$). This structure—termed a 'polar clasp' (*Gallagher-Jones et al., 2018*)—would be incompatible with interdigitation by opposing Q side chains (*Figure 2—figure supplement 2C*). We therefore quantified the frequencies and durations of side chain-side chain H-bonds in simulated singly N-substituted polyQ strands in explicit solvent, both in the presence and absence of restraints locking the backbone into a β-strand. This analysis showed no evidence that Q side chains preferentially H-bond adjacent N side chains (*Figure 2—figure supplement 2D–E*), ruling out the polar clasp mechanism of inhibition.

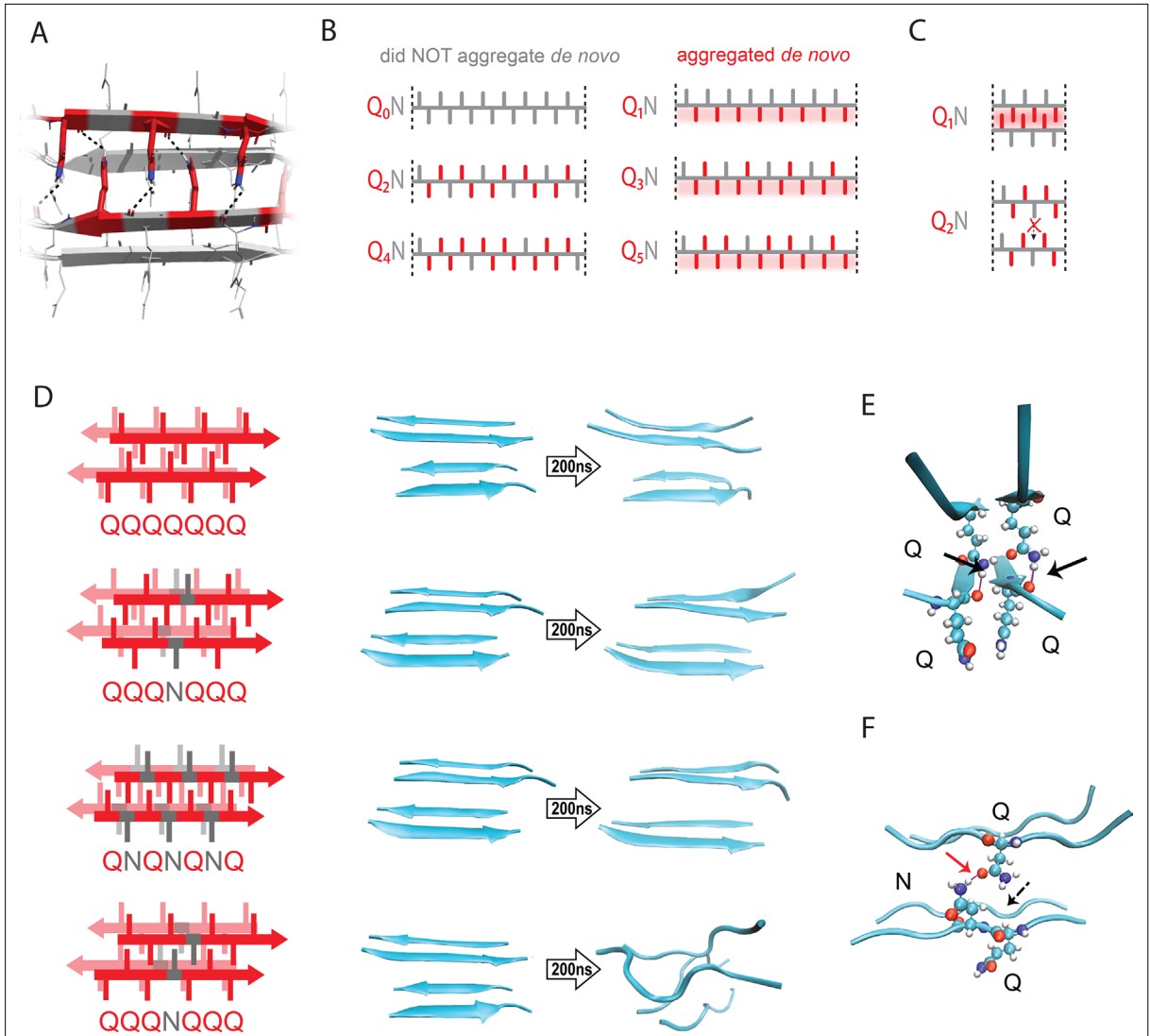

**Figure 2.** The pattern encodes a single unique steric zipper. (**A**) A view down the axis of a local segment of all-glutamine steric zipper, or "Q zipper", between two antiparallel two-stranded sheets. Residues with internally facing side-chains on the top layer are colored red to emphasize interdigitation and H-bonding (dashed lines) between the terminal amides and the opposing backbone. (**B**) Schema of the odd-even effect for Q zipper formation, showing side chain arrangements along a continuous β-strand for sequences composed of tandem repeats of Qs (red) interrupted by single Ns (gray). Shading highlights contiguous stretches of Qs that would occur in a continuous β-strand. Note that the illustrated strands will not necessarily be continuous in the context of the nucleus; i.e. the nucleus may contain shorter strands connected by loops. (**C**) Schema of the tertiary contacts between two β-strands, as in a steric zipper. The zipper can be formed only when the single interrupting non-Q residue follows an odd number of Qs (e.g. $Q_1N$), but not when it follows an even number of Qs (e.g. $Q_2N$). (**D**) Molecular simulations of model Q zippers formed by a pair of two-stranded antiparallel β-sheets, wherein non-Q residues (in red) face either inward or outward. The schema are oriented so the viewer is looking down the axis between two sheets. The zipper is stable for pure polyQ (QQQQQQQ, top simulation), or when substitutions face outward (QQQNQQQ, second simulation; and QNQNQNQ, fourth simulation), but not when even a single substitution faces inward (QQQNQQQ, third simulation). (**E**) Snapshot from the uninterrupted Q zipper simulation, showing H-bonds (black arrows) between internal extended Q side chains and the opposing backbones. (**F**) Snapshot from the internally interrupted Q zipper simulation, illustrating that the side chain of N is too short to H-bond the opposing backbone. However, the N side chain is long enough to H-bond the opposing Q side chain (red arrow), thereby intercepting the side chain-backbone H-bond that would otherwise occur (dashed arrow) between that Q side chain and the backbone amide adjacent to the N. This leads to dissolution of the zipper.

The online version of this article includes the following video and figure supplement(s) for figure 2:

**Figure supplement 1.** Molecular simulations of model steric zippers.

**Figure supplement 2.** Exploration of the mechanism by which N side chains disrupt Q zippers.

**Figure 2—video 1.** Simulation of the pure Q zipper shown in *Figure 2D* (top model), showing its persistence.

https://elifesciences.org/articles/86939/figures#fig2video1

*Figure 2 continued on next page*

*Figure 2 continued*

**Figure 2—video 2.** Simulation of the N-substituted Q zipper shown in *Figure 2D* (bottom model), showing the motions of side chains summarized in Figure S2C.

https://elifesciences.org/articles/86939/figures#fig2video2

In sum, our simulations are consistent with polyQ amyloid nucleating from a single Q zipper comprising two β-sheets engaged across an interface of fully interdigitated Q side chains.

## The Q zipper grows both laterally and axially

Prior structural studies show that polyQ amyloids have a lamellar or 'slab-like' architecture (*Boatz et al., 2020*; *Galaz-Montoya et al., 2021*; *Nazarov et al., 2022*; *Sathasivam et al., 2010*; *Sharma et al., 2005*). As a consequence of this structure, a hypothetical nucleus comprising a single Q zipper would have to propagate not only axially but also laterally as it matures toward amyloid, provided that both sides of each strand can form a Q zipper (*Figure 3A*). We observed that when expressed to high concentrations, sequences with $q$ values larger than 5 generally reached higher AmFRET values than those with $q$=3 or 5 (*Figure 3B*). This suggests that the former has a higher subunit density consistent with a multilamellar structure. We will therefore refer to such sequences as *bilaterally* contiguous or $Q_B$ and sequences that are only capable of Q zipper formation on one side of a strand, such as $Q_3X$ and $Q_5X$, as *unilaterally* contiguous or $Q_U$. AmFRET is a measure of total cellular FRET normalized by concentration. In a two phase regime, AmFRET scales with the fraction of protein in the assembled phase (*Posey et al., 2021*). At very high expression levels where approximately all the protein is assembled, AmFRET should theoretically approach a maximum value determined only by the proximity of fluorophores to one another in the assembled phase. This proximity in turn reflects the density of subunits in the phase as well as the orientation of the fluorophores on the subunits. To control for the latter, we varied the terminus and type of linker used to attach mEos3.1 to polyQ, $Q_3N$, and $Q_7N$. We found that polyQ and $Q_7N$ achieved higher AmFRET than $Q_3N$ regardless of linker terminus and identity (*Figure 3—figure supplement 1A*). The high AmFRET level achieved by polyQ amyloids therefore results from the subunits occurring in closer proximity than in amyloids that lack the ability to form lamella.

To more directly investigate physical differences between amyloids of these sequences, we analyzed the size distributions of SDS-insoluble multimers using semi-denaturing detergent-agarose gel electrophoresis (SDD-AGE) (*Halfmann and Lindquist, 2008*; *Kryndushkin et al., 2003*). We found that $Q_U$ amyloid particles were smaller than those of $Q_B$ (*Figure 3C*; *Figure 3—figure supplement 1B*). This difference necessarily means that they either nucleate more frequently (resulting in more but smaller multimers at steady state), grow slower, and/or fragment more than polyQ amyloids (*Knowles et al., 2009*). The DAmFRET data do not support the first possibility (*Figure 1—figure supplement 1D*). To compare growth rates, we examined AmFRET histograms through the bimodal region of DAmFRET plots for [$pin^-$] cells. Cells with intermediate AmFRET values (i.e. cells in which amyloid had nucleated but not yet reached steady state) were less frequent for $Q_U$ (*Figure 3D*; *Figure 3—figure supplement 1C*), which is inconsistent with the second possibility. Therefore, the SDD-AGE data are most consistent with the third possibility—increased fragmentation. We attribute this increase to a difference in the structures rather than sequences of the two types of amyloids, because Q and N residues are not directly recognized by protein disaggregases (*Alexandrov et al., 2012*; *Alexandrov et al., 2008*; *Osherovich et al., 2004*). Based on the fact that larger amyloid cores, such as those with multiple steric zippers, oppose fragmentation (*Tanaka et al., 2006*; *Verges et al., 2011*; *Zanjani et al., 2020*), these data therefore suggest that $Q_B$ amyloids have a lamellar (*Figure 3A*) architecture with more than two β-sheets.

## Q zippers poison themselves

A close look at the DAmFRET data reveals a complex relationship of amyloid formation to concentration. While all sequences formed amyloid more frequently with concentration in the low regime, as concentrations increased further, the frequency of amyloid formation plateaued somewhat before increasing again at higher concentrations (*Figure 4A*). A multiphasic dependence of amyloid formation on concentration implies that some higher-order species inhibits nucleation and/or growth (*Vitalis and Pappu, 2011*). The cells in the lower population lacked AmFRET altogether, suggesting that the

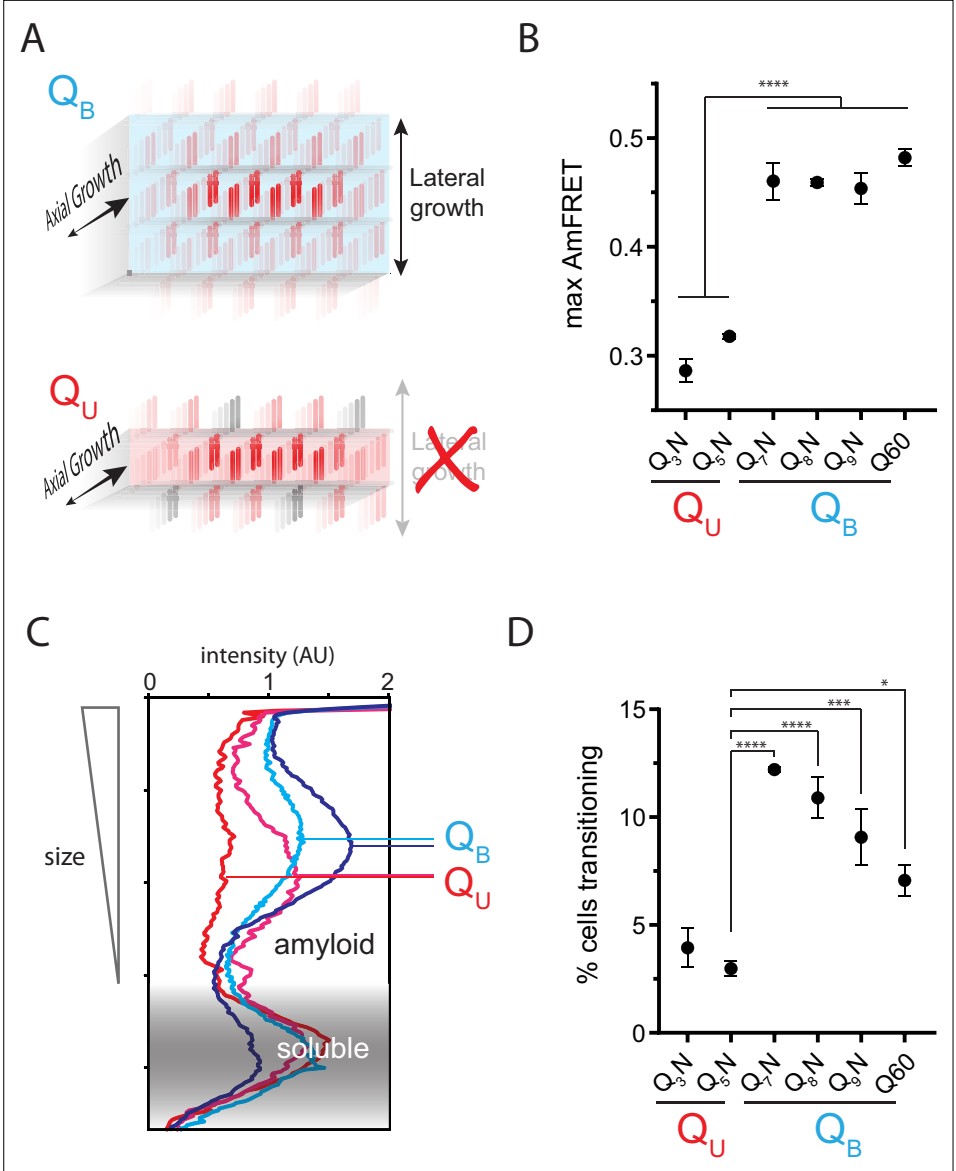

**Figure 3.** The Q zipper grows in two dimensions. (**A**) Schematic illustrating how sequences with bilaterally contiguous Qs (Q$_B$) can hypothetically allow for lateral growth (secondary nucleation) of Q zippers giving rise to lamellar amyloid fibers. In contrast, sequences with only unilateral contiguous Qs (Q$_U$) can form amyloids with only a single Q zipper. (**B**) Maximum AmFRET values for the indicated sequences in [*pin⁻*] cells, suggesting that Q$_B$ amyloids have a greater subunit density. Shown are means +/- SEM of the median AmFRET values of triplicates. **** p<0.0001; ANOVA and Dunnett's multiple comparison test. (**C**) Densitometric analysis of SDD-AGE characterizing amyloid length distributions for the indicated Q$_U$ and Q$_B$ amyloids, showing that Q$_B$ amyloid particles are larger. Data are representative of multiple experiments. (**D**) Fraction of cells at intermediate AmFRET values for the indicated sequences in [*pin⁻*] cells, suggesting that Q$_B$ amyloids grow slower. Shown are means +/- SEM of the percentage of cells between lower and upper populations, of triplicates. ****, ***, * p<0.0001,<0.001,<0.05; ANOVA.

The online version of this article includes the following figure supplement(s) for figure 3:

**Figure supplement 1.** Experiments supporting the existence of unilteral and bilateral Q zipper amyloids.

inhibitory species either have very low densities of the fluorophore (e.g. due to co-assembly with other proteins), and/or do not accumulate to detectable concentrations. Liquid-liquid phase separation by low-complexity sequences can give rise to condensates with low densities and heterogeneous compositions (*Wei et al., 2017*), and condensation has been shown to inhibit amyloid formation in

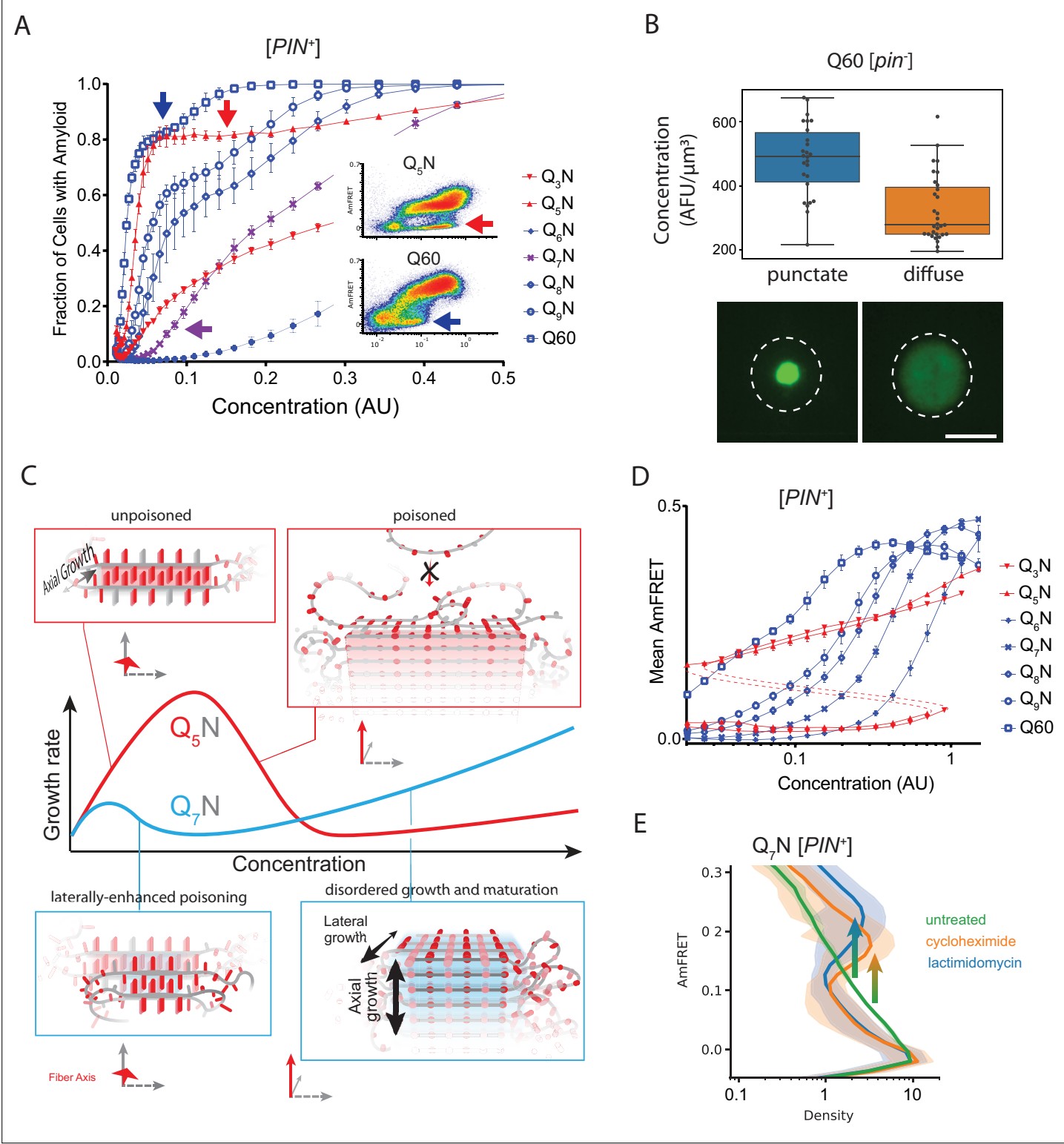

**Figure 4.** Q zippers poison themselves. (**A**) Fraction of [*PIN⁺*] cells in the AmFRET-positive population as a function of concentration for the indicated sequences. Arrows denote the population of cells with self-poisoned aggregation (inset), and the corresponding plateaus in the relationship of amyloid formation to concentration. The purple arrow highlights the sharp reduction in aggregation for $Q_7N$ relative to $Q_5N$, which we attribute to enhanced poisoning as a result of intramolecular Q zipper formation. Shown are means +/- SEM of triplicates. (**B**) Distribution of cytosolic concentrations (AFU/µm³) of Q60 in [*pin⁻*] cells either lacking or containing puncta, showing that the protein remains diffuse even when supersaturated relative to amyloid. Representative diffuse or punctate cells (N=31 and 26, respectively) of equivalent total concentration are shown. Scale bar: 5 µm. (**C**) Schematic

*Figure 4 continued on next page*

*Figure 4 continued*

illustrating self-poisoned growth as a function of concentration for Q zippers of $Q_5N$ and $Q_7N$. Conformational conversion of $Q_5N$ to amyloid decelerates (becomes poisoned) at high concentrations, as a consequence of polypeptides interfering with each other's conversion on the templating surface. This is illustrated here by the red trace and inset showing entangled, partially ordered polypeptides on the axial surface. The presence of contralaterally contiguous Qs in $Q_7N$ exacerbates poisoning at low concentrations, as illustrated here by the blue trace and inset showing partially-ordered species immobilized with bilateral zippers. Growth resumes at high concentrations with the addition of successive zippers. (**D**) Graph of spline fits of AmFRET values for the indicated sequences in [*PIN*⁺] cells. The upper and lower populations of $Q_3N$ and $Q_5N$ were treated separately due to the extreme persistence of the low population for these sequences. The red dashed lines denote these are subpopulations of the same samples. The ability of amyloid to grow at low concentrations fell sharply with the onset of bilateral contiguity at $Q_6N$ and then gradually increased with higher *q* values. Shown are means +/- SEM of triplicates. (**E**) Histogram of AmFRET values for $Q_7N$ -expressing cells transitioning from the low to high populations (boxed region from DAmFRET plot in *Figure 4—figure supplement 1D*) upon translation inhibition for six hours following 18 hours of expression. Shown are means +/- 95% CI of biological triplicates. Blocking new protein synthesis prior to analysis causes AmFRET to rise, whether by cycloheximide or lactimidomycin (p<0.01,<0.05, respectively, Dunnett's test).

The online version of this article includes the following figure supplement(s) for figure 4:

**Figure supplement 1.** Experiments supporting the self-poisoning mechanism.

some contexts (*Gabryelczyk et al., 2022*; *Küffner et al., 2021*; *Lipiński et al., 2022*). We therefore inspected the subcellular distribution of representative proteins in different regions of their respective DAmFRET plots. The amyloid-containing cells had large round or stellate puncta, as expected. To our surprise, however, cells lacking AmFRET contained exclusively diffuse protein (no detectable puncta), even at high expression (*Figure 4B*; *Figure 4—figure supplement 1A*). This means that naive polyQ does not itself phase separate under normal cellular conditions. The inhibitory species are therefore some form of soluble oligomer. That these oligomers are too sparse to detect by AmFRET, even at high total protein concentration, further suggests that their formation is kinetically limited, and, therefore, subsequent to Q zipper formation. In other words, Q zipper nuclei appear to form dead-end (kinetically trapped) oligomers when the concentration of soluble polyQ is too high.

We were struck by the similarity of this phenomenon to 'self-poisoned' crystallization of long organic polymers in solution (*Organ et al., 1989*; *Sadler, 1983*; *Ungar et al., 2005*; *Whitelam et al., 2016*; *Zhang et al., 2016a*; *Zhang et al., 2021*). Self-poisoning is a deceleration of crystal growth with increasing driving force. In short, when multiple molecules simultaneously engage the templating crystal surface, they tend to 'trap' one another in partially ordered configurations that block the recruitment of subsequent molecules (*Figure 4C*, red). This phenomenon requires conformational conversion on the templating surface to be slow relative to the arrival of new molecules. That polyQ amyloid formation may be susceptible to self-poisoning is consistent with its extremely viscous dynamics (*Crick et al., 2006*; *Dougan et al., 2009*; *Kang et al., 2017*; *Walters and Murphy, 2009*) that has already been shown to limit the rate of amyloid elongation (*Walters et al., 2012*).

Self-poisoning is most extreme where phase boundaries converge, as when a polymer is equally compatible with either of two crystal polymorphs (*Hu, 2018*; *Ungar et al., 2005*). For example, the rate of polymer crystallization rises, falls, and then rises again as a function of polymer length, where the second minimum results from the polymer heterogeneously conforming to either a single-long-strand polymorph or a hairpin polymorph (*Figure 4—figure supplement 1B*). This is a consequence of the secondary nucleation of ordering *within* the molecule after it engages the templating surface (*Hu, 2018*; *Ungar et al., 2005*; *Zhang et al., 2016a*; *Zhang et al., 2021*). We detected an analogous phenomenon for Q zippers. Specifically, we noticed that $Q_5N$ but not $Q_7N$ formed amyloid robustly at low concentrations (*Figure 4A*, purple arrow). We were initially perplexed by this because both sequences have a fully contiguous pattern of Qs at every other position that should allow for Q zipper nucleation. We then realized, however, that the anomalous solubility of $Q_7N$ coincides with the expected enhancement of self-poisoning at the transition between single long Q zippers (favored by $Q_U$ sequences) and short lamellar Q zippers (favored by $Q_B$ sequences). In other words, the difference between $Q_5N$ and $Q_7N$ can be explained by the latter forming *intramolecular* Q zippers with strands only six residues long, *contralaterally* to the intermolecular Q zipper interface (*Figure 4C*, blue). Here, we use the term 'contralateral' to refer to the side chains of a β-strand on the other side of the backbone from those in the intermolecular Q zipper. Consistent with the behaviors of other self-poisoning polymers, $Q_B$ sequences increasingly escaped the poisoned state as the potential length of lamellar strands increased beyond the minimum, i.e. with increasing *q* values beyond 6 (*Figure 4D*).

In our living experimental system, ongoing protein translation will cause the polypeptide to become increasingly supersaturated with respect to the self-poisoned amyloid phase. However, as concentrations increase, the contributions of unstructured low affinity interactions (primarily H-bonds between Q side chains *Kang et al., 2018*; *Punihaole et al., 2018*) to intermolecular associations will increase relative to the contributions of Q zipper elements. Disordered polyQ has a greater affinity for itself than does disordered polyN (*Halfmann et al., 2011*). The ability of $Q_B$ sequences to escape poisoning with increasing concentrations may therefore be a simple consequence of their greater content of Qs. Indeed, polyQ amyloid (*Walters et al., 2012*), as for other polymer crystals (*Zhang et al., 2021*), has a relatively disordered growth front at high polymer concentrations.

Our finding that polyQ does not phase-separate prior to Q zipper nucleation implies that intermolecular Q zippers will be less stable than intramolecular Q zippers of the same length, because the effective concentration of Q zipper-forming segments will always be lower outside the intramolecular globule than inside it. Therefore, the critical strand length for growth of a single (non-lamellar) Q zipper must be longer than six, whereas strands of length six will only grow in the context of lamella. To test this prediction, we designed a series of sequences with variably restricted unilateral or bilateral contiguity (*Figure 4—figure supplement 1C–D*). We found that, while all sequences formed amyloid with a detectable frequency in [*PIN⁺*] cells, only those with at least five unilaterally contiguous Qs, or at least six bilaterally contiguous Qs, did so in [*pin⁻*] cells. Given that the ability to nucleate in the absence of a pre-existing conformational template is characteristic of Q zippers, this result is consistent with a threshold strand length of nine or ten residues for single Q zippers to propagate, and six residues for lamellar Q zippers to propagate.

One consequence of self-poisoning is that crystallization rates accelerate as the concentration of monomers fall due to their (initially slow) deposition onto the crystal (*Ungar et al., 2005*). In our system, poisoning should be most severe in the early stages of amyloid formation when templates are too few for polypeptide deposition to outpace polypeptide synthesis, thereby maintaining concentrations at poisoning levels. For sufficiently fast nucleation rates and slow growth rates, this property predicts a bifurcation in AmFRET values as concentrations enter the poisoning regime, as we see for $Q_B$ in [*PIN⁺*] cells. To determine if the mid-AmFRET cells in the bifurcated regime indeed have not yet reached steady state, we used translation inhibitors to stop the influx of new polypeptides—and thereby relax self-poisoning—for 6 hr prior to analysis. As predicted, treated AmFRET-positive cells in the bifurcated regime achieved higher AmFRET values than cells whose translation was not arrested (*Figure 4E*; *Figure 4—figure supplement 1E*).

## The nucleus forms within a single molecule

Because amyloid formation is a transition in both conformational ordering *and* density, and the latter is not rate-limiting over our experimental time-scales (*Khan et al., 2018*), the critical conformational fluctuation for most amyloid-forming sequences is generally presumed to occur within disordered multimers (*Auer et al., 2008*; *Buell, 2017*; *Serio et al., 2000*; *Vekilov, 2012*; *Vitalis and Pappu, 2011*). In contrast, homopolymer crystals nucleate preferentially within monomers when the chain exceeds a threshold length (*Hu, 2018*; *Organ et al., 1989*; *Ungar et al., 2005*; *Xu et al., 2021*; *Zhang et al., 2018*). This follows from the tendency of ordering to arrest via self-poisoning at high local concentrations, which effectively shifts the rate-limiting step to the growth of a higher order relatively disordered species. The possibility that polyQ amyloid may nucleate as a monomer has long been suspected to underlie the length threshold for polyQ pathology (*Chen et al., 2002*). A minimal intramolecular Q zipper would comprise a pair of two-stranded β-sheets connected by loops of approximately four residues (*Chou and Fasman, 1977*; *Hennetin et al., 2006*). If we take the clinical threshold of approximately 36 residues as the length required to form this structure, then the strands must be approximately six residues with three unilaterally contiguous Qs—exactly the length we deduced for intramolecular strand formation in the previous section.

This interpretation leads to an important prediction concerning the inability of $Q_4N$ to form amyloid. With segments of four unilaterally and bilaterally contiguous Qs, this sequence should be able to form the hypothetical intramolecular Q zipper, but fail to propagate it either axially or laterally as doing so requires at least five unilaterally or six bilaterally contiguous Qs, respectively. If the rate-limiting step for amyloid is the formation of a single short Q zipper, then appending $Q_4N$ to another Q zipper amyloid-forming sequence will facilitate its nucleation. If nucleation instead requires longer or

lamellar zippers, then appending $Q_4N$ will inhibit nucleation. To test this prediction, we employed $Q_3N$ as a non-lamellar 'sensor' of otherwise cryptic Q zipper strands donated by a 30 residue $Q_4N$ tract appended to it. To control for the increased total length of the polypeptide, we separately appended a non Q zipper-compatible tract: $Q_2N$. As predicted, the $Q_4N$ appendage increased the fraction of cells in the high-AmFRET population relative to those expressing $Q_3N$ alone, and even more so relative to those expressing the $Q_2N$-appended protein (*Figure 5A*; *Figure 5—figure supplement 1A*). These data are consistent with a single short Q zipper as the amyloid nucleus.

To now determine if nucleation occurs preferentially in a single polypeptide chain, we genetically fused $Q_U$ and $Q_B$ to either of two homo-oligomeric modules: a coiled coil dimer (oDi, *Fletcher et al., 2012*), or a 24-mer (human FTH1, *Bracha et al., 2018*). Importantly, neither of these fusion partners contain β-structure that could conceivably template Q zippers. Remarkably, the oDi fusion reduced amyloid formation, and the FTH1 fusion all but eliminated it, for both $Q_U$ and $Q_B$ sequences (*Figure 5B*; *Figure 5—figure supplement 1B*). To exclude trivial explanations for this result, we additionally tested fusions to oDi on the opposite terminus, with a different linker, or with a mutation designed to block dimerization (designated Odi{X}). The inhibitory effect of oDi manifested regardless of the terminus tagged or the linker used, and the inhibitory effect was suppressed by the monomerizing mutation (*Figure 5—figure supplement 1C–F*), altogether confirming that preemptive oligomerization inhibits amyloid nucleation. Whether it does so by destabilizing the intramolecular Q zipper, or instead by decelerating a subsequent step in its maturation to amyloid—such that that step becomes rate-limiting (and therefore the new nucleus)—remains to be determined.

Finally, we designed a sequence representing the minimal 'polyQ' amyloid nucleus. This sequence has the precise number and placement of Qs necessary for amyloid to form via intramolecular Q zipper nucleation: four tracts of six Qs linked by minimal loops of three glycines. The sequence itself is too short—at 33 residues—to form amyloid unless the glycine loops are correctly positioned (recall from *Figure 1D* and *Figure 1—figure supplement 1A* that pure polyQ shorter than 40 residues does not detectably aggregate). We found that the protein indeed formed amyloid robustly, and with a concentration-dependence and [*PIN*⁺]-independence that is characteristic of $Q_B$ (*Figure 5C*, *Figure 5—figure supplement 1G*). If the nucleus is indeed a monomer, then a defect in even one of the four strands should severely restrict amyloid formation. We therefore mutated a single Q residue to an N. Remarkably, this tiny change—removing just one carbon atom from the polypeptide—completely eliminated amyloid formation (*Figure 5C*, *Figure 5—figure supplement 1G*). We conclude that the polyQ amyloid nucleus is an intramolecular Q zipper.

## Discussion
### PolyQ amyloid begins *within* a molecule

The initiating molecular event leading to pathogenic aggregates is the most important yet intractable step in the progression of age-associated neurodegenerative diseases such as Huntington's. Here, we used our recently developed assay to identify sequence features that govern the dependence of nucleation frequency on concentration and conformational templates. Doing so revealed a pattern of Q residues (specifically, $(QXQXQX_{>3})4$ where $X_{>3}$ denotes any length of three or more residues of any composition) governing the concentration-dependence of amyloid formation, which in turn allowed us to delineate the critical lengths and numbers of β-strands in the nucleus. Collectively our data demonstrate that amyloid formation by pathologically expanded polyQ begins with the formation of a minimal steric zipper of interdigitated side chains—a 'Q zipper'—within a single polypeptide molecule. In this respect, amyloid formation by polyQ resembles the crystallization of long synthetic homopolymers in dilute solution (*Hu, 2018*; *Hu et al., 2003*; *Lauritzen and Hoffman, 1960*).

Kinetic studies performed in vitro (*Bhattacharyya et al., 2005*; *Chen et al., 2002*; *Kar et al., 2011*) and in neuronal cell culture *Colby et al., 2006* have found that polyQ aggregates with a reaction order of approximately one. A recent study investigating the aggregation kinetics of Q40-YFP in *C. elegans* body wall muscle cells observed a reaction order of 1.6 (*Sinnige et al., 2021*), which is slightly higher than 1 perhaps as a consequence of their employing a weakly homodimeric (*Landgraf et al., 2012*; *Snapp et al., 2003*) fluorescent protein fusion and positive charges flanking the Q tract. The latter may be incompatible with a four stranded intramolecular Q zipper because the beginning and end of the Q tract necessarily occur in close proximity and would therefore be destabilized by charge

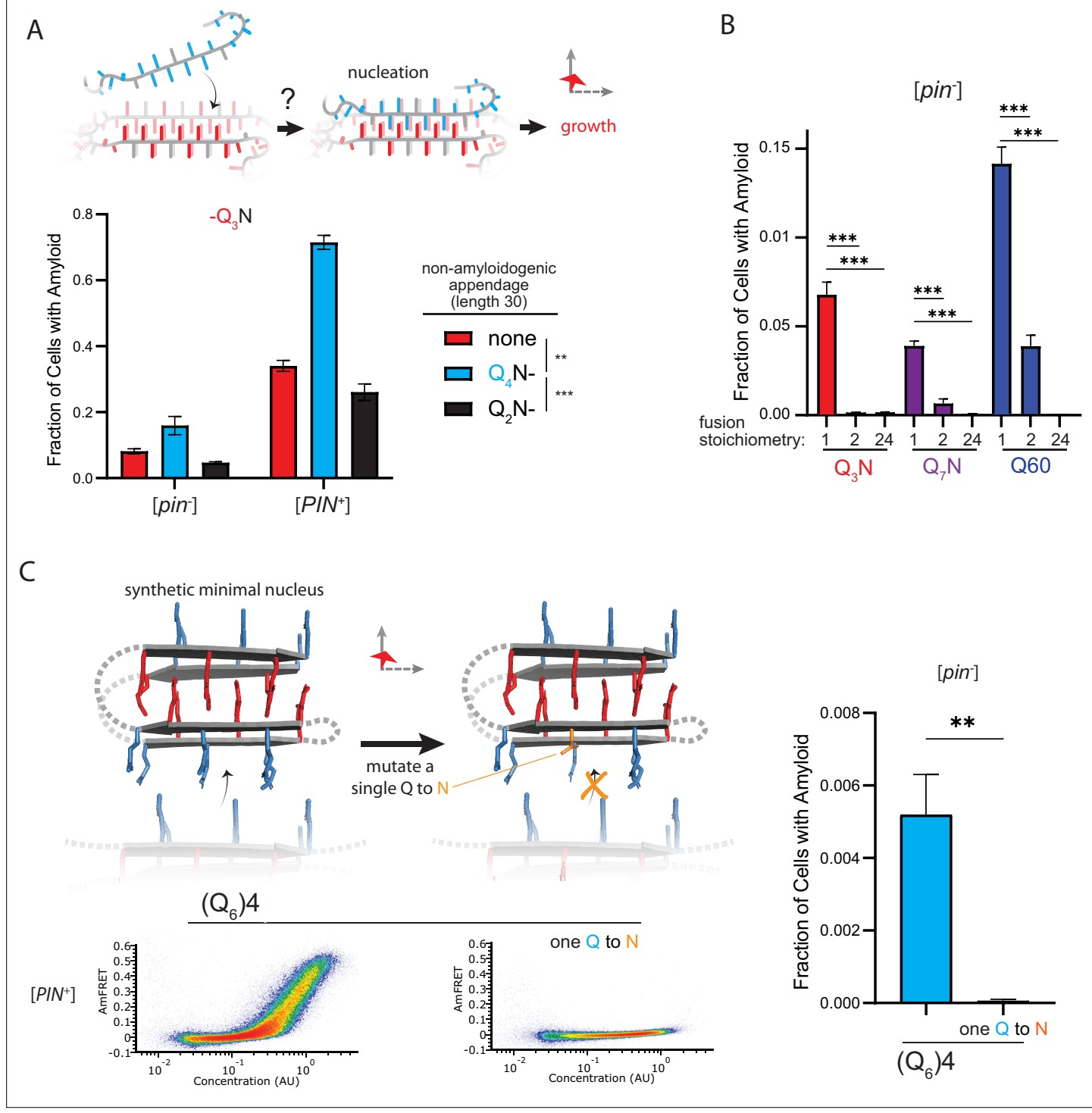

**Figure 5.** The nucleus forms within a single molecule. (**A**) Fraction of cells in the AmFRET-positive population when expressing $Q_3N$ with the indicated non-amyloidogenic fusions. Shown are means +/- SEM of triplicates. ***, ** p<0.001,<0.01; ANOVA. (**B**) Fraction of cells in the AmFRET-positive population (with higher AmFRET than that of the oligomer itself) when expressing the indicated protein fused to proteins with the indicated stoichiometry. Shown are means +/- SEM of triplicates. *** p<0.001; t-test. (**C**) Schema, DAmFRET plots, and quantitation of amyloid formation by a synthetic minimal polyQ amyloid-forming sequence. Q side chains in the nucleating zipper are colored red, while those necessary for growth of the zipper—which requires lateral propagation due to its short length—are colored blue. The three G3 loops are represented by dashed gray lines; the actual topology of the loops may differ. Mutating a single Q to N blocks amyloid formation. Shown are means +/- SEM of triplicates. ** p<0.01; t-test.

The online version of this article includes the following figure supplement(s) for figure 5:

**Figure supplement 1.** Experiments supporting monomeric nucleation.

repulsion. If one assumes that pathogenic polyQ nucleates homogeneously, all of these studies imply that it does so as a monomer. If polyQ instead nucleates heterogeneously, as has been interpreted from observations that it forms apparently non-amyloid multimers prior to amyloids in vitro (*Crick et al., 2013*; *Posey et al., 2018*), then the reaction order of 1 instead suggests that the rate-limiting conformational event occurs within a single molecule inside a pre-amyloid multimer (*Šarić et al., 2016*). Importantly, both possibilities imply that the critical conformational features of the amyloid nucleus involve only one polypeptide molecule (or perhaps its discontinuous equivalent, in the latter case). By explicitly manipulating concentration, length, density and conformational heterogeneities, we confirm that the rate-limiting step for pathologic polyQ amyloid formation does indeed occur within a monomer. Moreover, we show that it does so even in the dynamic, intermingled, living cellular environment.

Once formed, this minimal Q zipper germinates in all three dimensions to ultimately produce amyloid fibers with lamellar zippers longer than that of the nucleus proposed here. Both features have been confirmed by structural studies (*Boatz et al., 2020*; *Galaz-Montoya et al., 2021*; *Hoop et al., 2016*; *Nazarov et al., 2022*). In the case of polyQ, the short-stranded nucleus is a catalyst for longer, lamellar Q zippers that make up the amyloid core. That the nascent amyloid should become progressively more ordered is in keeping with decades of experimental and theoretical work by polymer physicists that establishes a thermodynamic drive toward strand lengthening in crystals (*Flory, 1962*; *Keller and O'connor, 1957*; *Lauritzen and Hoffman, 1960*; *Phan and Schmit, 2020*; *Xu et al., 2021*; *Zhang et al., 2016a*). The minimally competent zipper identified here provides lower bounds on the widths of fibers—they can be as thin as two sheets (1.6 nm) and as short as six residues. This hypothetical restriction is thinner and shorter than the cores of all known amyloid structures of full length polypeptides (*Sawaya et al., 2021*). Remarkably, observations from recent cryoEM tomography studies closely match this prediction: pathologic polyQ or huntingtin amyloids, whether in cells or in vitro, feature a slab-like architecture with restrictions as thin as 2 nm (*Galaz-Montoya et al., 2021*) and as short as five residues (*Nazarov et al., 2022*).

Multidimensional polymer crystal growth offers a simple explanation for the characteristically antiparallel arrangement of strands in polyQ amyloid. Each lamellum first nucleates then propagates along the axial surface of the templating fiber by the back-and-forth folding of monomers (*Zhang et al., 2016a*). In the context of polyQ, the end result is a stack of β-hairpins. If lamella are responsible for antiparallel strands, then $Q_U$ sequences should form typical amyloid fibers with parallel β-strands. The only atomic resolution structure of a Q zipper amyloid happens to be of a protein with striking unilateral (but not bilateral) contiguity (*Hervas et al., 2020*), and the strands are parallel as predicted.

The flanking regions and other features of full-length proteins have been shown to modulate both the aggregation and toxicity of polyQ (*Adegbuyiro et al., 2017*; *Arndt et al., 2020*; *Ceccon et al., 2022*; *Elena-Real et al., 2023a*; *Elena-Real et al., 2023b*; *Hong et al., 2019*; *Khaled et al., 2023*; *Kuiper et al., 2017*; *Lieberman et al., 2019*; *Silva et al., 2018*). The extent to which our findings will translate in these different contexts remains to be determined. Nevertheless, that the intrinsic behavior of the polyQ tract itself is central to pathology is evident from the fact that the nine pathologic polyQ proteins have similar length thresholds despite different functions, flanking domains, interaction partners, and expression levels.

Only two of the nine polyQ diseases have length thresholds below the minimum length for an intramolecular Q zipper as deduced here (*Lieberman et al., 2019*). These two proteins—SCA2 (32 residues) and SCA6 (21 residues)—also appear to be atypical for this class of diseases in that disease onset is accelerated in homozygous individuals (*Laffita-Mesa et al., 2012*; *Mariotti et al., 2001*; *Soga et al., 2017*; *Spadafora et al., 2007*; *Tojima et al., 2018*). This is consistent with nucleation occurring in oligomers, as would be necessary for polyQ tracts that are too short to form all four strands of the minimal Q zipper within one molecule.

We expect that the monomeric nucleus of polyQ will prove to be unusual among pathologic amyloid-forming proteins. Among the hundreds of amyloid-forming proteins we have now characterized by DAmFRET (*Khan et al., 2018*; *Posey et al., 2021*, and unpublished), only Cyc8 PrD has a similar concentration-dependence. We now attribute this to self-poisoning due to its exceptionally long tract of 50 unilaterally contiguous Qs, primarily in the form of QA dipeptide repeats.

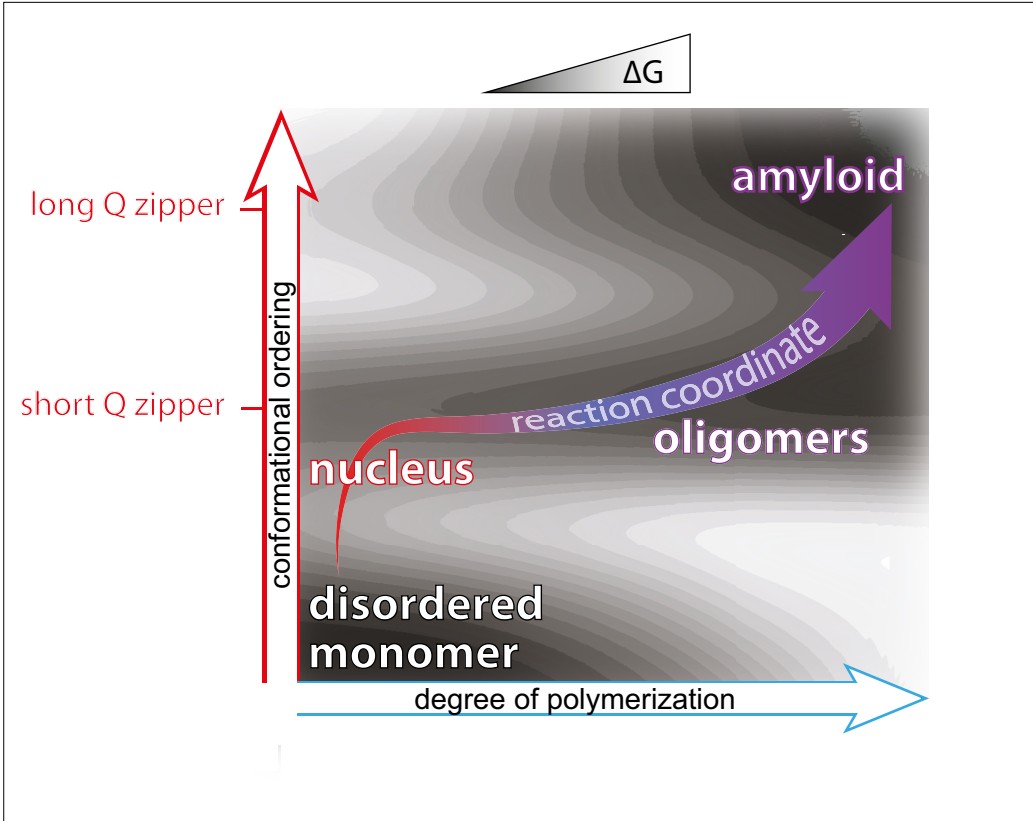

**Figure 6.** Summary of aggregation mechanism. Schematic of the free energy landscape for amyloid formation by pathologically expanded polyQ at approximately physiological concentrations, showing the reaction pathway as a function of the conformational ordering and degree of polymerization of the species. Qualitative topological features of the landscape, but not absolute heights and positions, are as deduced herein. Naive monomers exist in a local energy minimum at maximum disorder, while mature amyloid exists in a global energy minimum with long lamellar Q zippers. The middle and upper horizontal basins represent Q zippers with short (~6 residue) and long (~11 residue) strands, respectively. Nucleation occurs with the formation of a short intramolecular Q zipper, which then oligomerizes via axial and lateral recruitment of other polypeptides. The Q zipper eventually lengthens, allowing for the growth of mature amyloid.

## Proteotoxic multimers may be self-poisoned polymer crystals

Soluble oligomers accumulate during the aggregation of pathologically lengthened polyQ and/or Htt in vitro, in cultured cells, and in the brains of patients (*Auer et al., 2008*; *Hsieh et al., 2017*; *Legleiter et al., 2010*; *Levin et al., 2014*; *Liang et al., 2018*; *Li et al., 2010*; *Olshina et al., 2010*; *Sathasivam et al., 2010*; *Sil et al., 2018*; *Takahashi et al., 2008*; *Vitalis and Pappu, 2011*; *Yamaguchi et al., 2005*; *Zanjani et al., 2020*). They are likely culprits of proteotoxicity (*Kim et al., 2016*; *Leitman et al., 2013*; *Lu and Palacino, 2013*; Matlahov and *Matlahov and van der Wel, 2019*; *Takahashi et al., 2008*; *Wetzel, 2020*).

We summarize our findings on the relationship of heterogeneities to polyQ amyloid formation using an energy landscape to illustrate the pathway of polyQ aggregation in terms of both intra-molecular (conformational) ordering and degree of polymerization (*Figure 6*). We did not observe disordered oligomerization in cells. Instead, all forms of multimerization appeared to be dependent on Q zipper nucleation. That kinetically arrested aggregates emerge from the same event that limits the rate of amyloid formation suggests a resolution to the paradox that polyQ diseases are character-ized by amyloid formation despite the implication of non-amyloid species (*Kim et al., 2016*; *Leitman et al., 2013*; *Lu and Palacino, 2013*; Matlahov and *Matlahov and van der Wel, 2019*; *Takahashi et al., 2008*; *Wetzel, 2020*). The decades-long course of clinical disease provides an upper limit on the timescale of candidate proteopathic events. Because of this long timescale, the event of Q zipper nucleation—even if it occurs as rarely as once per neuron over the lifetime of a patient—could in

principle dominate the kinetics of disease. Likewise, a slow-growing self-poisoned Q zipper may very well take years to kill the neuron. This consideration has important implications for disease models, and specifically cautions against the use of overexpression to accelerate proteotoxicity. By circumventing the self-poisoned state, overexpression plausibly changes the nature of the toxic species and triggers pathways of cell death that may be irrelevant to disease. From our perspective, the ideal model for polyQ diseases will accelerate Q zipper nucleation without changing amyloid equilibrium, as could be achieved by expressing a synthetic Q zipper template.

Notwithstanding one report to the contrary (*Peskett et al., 2018*)—which we respectfully attribute to a known artifact of the FLAG tag when fused to polyQ (*Duennwald et al., 2006a*; *Duennwald et al., 2006b*; *Jiang et al., 2017*)—our findings corroborate prior demonstrations that polyQ (absent flanking domains) does not phase separate prior to amyloid formation in cells, whether expressed in human neuronal cells (*Colby et al., 2006*; *Kakkar et al., 2016*), *C. elegans* body wall muscle cells (*Sinnige et al., 2021*), or [*pin*⁻] yeast cells (*Duennwald et al., 2006a*; *Jiang et al., 2017*). In contrast, full-length Huntingtin protein with a pathologically lengthened polyQ tract does phase separate in cells (*Aktar et al., 2019*; *de Mattos et al., 2022*; *Peskett et al., 2018*; *Wan et al., 2021*). In light of our discovery that polyQ amyloid formation is blocked by oligomerization, this property suggests a simple explanation for the otherwise paradoxical fact that Huntington's Disease onset and severity do not increase in homozygous individuals (*Cubo et al., 2019*; *Lee et al., 2019*; *Wexler et al., 1987*). Specifically, if mutant Huntingtin forms an endogenous condensate (if only in some microcompartment of the cell) when expressed from just one allele, then the buffering effect of phase separation (*Klosin et al., 2020*) will prevent a second allele from increasing the concentration of nucleation-competent monomers.

Our data hint at the nature of these arrested multimers. For $Q_U$, the multimers failed to accumulate to a level detectable by AmFRET, and failed to coalesce to microscopic puncta. They therefore seem to involve only a small fraction of the total protein, presumably limited to soluble species that have acquired a nascent Q zipper. The disordered polyQ globule is extremely viscous and this causes very slow conformational conversion on the tips of growing fibers (*Bhattacharyya et al., 2005*; *Walters et al., 2012*). It should therefore be highly susceptible to self-poisoning as has been widely observed and studied for polymer crystals in vitro (*Jiang et al., 2016*; *Ungar and Keller, 1987*; *Whitelam et al., 2016*; *Zhang et al., 2020*; *Zhang et al., 2018*). In short, the multimers are nascent amyloids that templated their own arrest in the earliest stages of growth following nucleation.

Our observations that arrested post-nucleated species commence growth at higher concentrations echoes in vitro findings by Pappu and colleagues that polyQ populates a soluble phase of large spherical oligomers at concentrations that are slightly subsaturated with respect to fibrils (*Crick et al., 2013*; *Posey et al., 2018*). These had been postulated to form via liquid-like disordered interactions. We find that interpretation to be incompatible with polyQ amyloids as we know them because it seems to preclude a driving force for internal structure. We therefore offer an alternative interpretation of these large oligomers, namely, as the partially ordered products of self-poisoned nucleation. The existence of a lower phase boundary for these species is consistent with our findings that intra- and intermolecular Q zippers have different critical lengths, which follows from the fact that intramolecular segments are covalently constrained to much higher effective concentrations that allow for intramolecular β-sheets to have shorter strands. The upper phase boundary (above which fibrils appear) reflects the resumption of growth via relatively low-affinity disordered intermolecular interactions. If our interpretation of the oligomeric phase is correct, we can expect structural analyses to uncover amyloid-like internal structure, and perhaps also a gross resemblance to semicrystalline spherulites that have been described for other polymer crystal and amyloid-forming systems such as polyglutamic acid (*Crist and Schultz, 2016*; *Hu, 2018*; *Stehli et al., 2015*; *Vetri and Foderà, 2015*). Consistent with this expectation, a recent study using cryogenic nanoprobe small angle X-ray scattering to probe the structure of unlabeled mutant Htt exon 1 in cells revealed a population of soluble oligomers with features of stacked β-sheets (*Rumancev et al., 2021*).

We do not yet know if these aborted amyloids contribute to pathology. However, the role of contralateral zippers in their formation is consistent with the apparent protective effect of CAT trinucleotide insertions that preserve unilateral contiguity (QHQHQH) in the polyQ disease protein, SCA1 (*Menon et al., 2013*; *Nethisinghe et al., 2018*; *Sen et al., 2003*), and the absence of bilateral contiguity in

the only known functional Q zipper—that of the neuronal translational regulator, Orb2 (*Hervas et al., 2020*). We speculate that the toxicity of polyQ amyloid arises from its lamellar architecture.

## Concluding remarks

The etiology of polyQ pathology has been elusive. Decades-long efforts by many labs have revealed precise measurements of polyQ aggregation kinetics, atomistic details of the amyloid structure, a catalog of proteotoxic candidates, and snapshots of the conformational preferences of disordered polyQ. What they have not yet led to are treatments. We synthesized those insights to recognize that pathogenesis likely begins with a very specific conformational fluctuation. We set out to characterize the nature of that event in the cellular milieu, deploying a technique we developed to do just that, and arrived at an intramolecular four-stranded polymer crystal. Our findings rationalize key aspects of polyQ diseases, such as length thresholds, kinetics of progression, and involvement of pre-amyloid multimers. More importantly, they illuminate a new avenue for potential treatments. Current therapeutic efforts focused on lowering the levels of mutant Huntingtin have not been successful. As an admittedly radical alternative, we suggest that therapies designed to (further) oligomerize huntingtin *preemptively* will delay nucleation and thereby decelerate the disease.

## Materials and methods

### Plasmid and strain construction

ORFs were codon optimized for expression in *S. cerevisiae*, synthesized, and cloned into vector V08 *Khan et al., 2018* by Genscript (Piscataway, NJ, USA). See *Supplementary file 1* for all full-length protein sequences.

Yeast strain rhy3078a ([*PIN*⁺]) was constructed as follows. We first deleted *PDR5* and *ATG8* from strain rhy1713 *Khan et al., 2018* by sequentially mating and sporulating it with the respective strains from the *MATa* deletion collection (Open Biosystems). We then used PCR-based mutagenesis (*Goldstein and McCusker, 1999*) with template vector CX (Miller et al. submitted) to integrate BDFP1.6:1.6 prior to the stop codon of chromosomal *PGK1*. Yeast strain rhy3082 ([*pin*⁻]) is rhy3078a with the amyloid form of Rnq1 eliminated by passaging it four times on YPD plates containing 3 mM GdHCl, a prion-curing agent (*Ferreira et al., 2001*).

### DAmFRET

The yeast strains were transformed using a standard lithium acetate protocol with plasmids encoding the sequence to be tested as a fusion to the indicated tags (*Supplementary file 1*) under the control of the *GAL1* promoter.

Individual colonies were picked and incubated in 200 µL of a standard synthetic media containing 2% dextrose (SD -ura) overnight while shaking on a Heidolph Titramax-1000 at 1000 rpm at 30 °C. Following overnight growth, cells were spun down and resuspended in a synthetic induction media containing 2% galactose (SGal -ura). Cells were induced for 16 hr while shaking before being resuspended in fresh 2% SGal -ura for 4 hr to reduce autofluorescence. A total of 75 µL of cells were then re-arrayed from a 96-well plate to a 384-well plate and photoconverted, while shaking at 800 rpm, for 25 min using an OmniCure S1000 fitted with a 320–500 nm (violet) filter and a beam collimator (Exfo), positioned 45 cm above the plate.

High-throughput flow cytometry (10 µL/well, flow speed 1.8) was performed on the Bio-Rad ZE5 with a Propel automation setup and a GX robot (PAA Inc). Donor and FRET signals were collected from a 488 nm laser set to 100 mW, with voltages 351 and 370 respectively, into 525/35 and 593/52 detectors. Acceptor signal was collected from a 561 nm laser set to 50 mW with voltage at 525, into a 589/15 detector. Autofluorescence was collected from a 405 nm laser at 100 mW and voltage 450, into a 460/22 detector.

Compensation was performed manually, collecting files for non-photoconverted mEos3.1 for pure donor fluorescence and dsRed2 to represent acceptor signal as it has a similar spectrum to the red form of mEos3.1. FRET is compensated only in the direction of donor and acceptor fluorescence out of FRET channels, as there is not a pure FRET control. Acceptor fluorescence intensities were divided by side scatter (SSC), a proxy for cell volume (Miller et al. submitted), to convert them to concentration (in arbitrary units).

Imaging flow cytometry was conducted as in previous work (*Khan et al., 2018*; *Venkatesan et al., 2019*).

## DAmFRET automated analysis

FCS 3.1 files resulting from assay were gated using an automated R-script running in flowCore. Prior to gating, the forward scatter (FS00.A, FS00.W, FS00.H), side scatter (SS02.A), donor fluorescence (FL03.A) and autofluorescence (FL17.A) channels were transformed using a logicle transform in R. Gating was then done using flowCore by sequentially gating for cells using FS00.A vs SS02.A then selecting for single cells using FS00.H vs FS00.W and finally selecting for expressing cells using FL03.A vs FL17.A.

Gating for cells was done using a rectangular gate with values of Xmin = 2.7, Xmax = 4.8, Ymin = 2.7, Ymax = 4.7. Gating for single cells was done using a rectangular gate with values of Xmin = 4.45, Xmax = 4.58, Ymin = 2.5, Ymax = 4.4. Gating of expressing cells was done using a polygon gate with x/y vertices of (1,0.1), (1.8, 2), (5, 2), (5,0.1). Cells falling within all of these gates were then exported as FCS3.0 files for further analysis.

The FCS files resulting from the autonomous gating in Step 1 were then utilized for the JAVA-based quantification of a curve similar to the analysis found in the original assay (*Khan et al., 2018*). Specifically:

The quantification procedure first divides a defined negative control DAmFRET histogram into 64 logarithmically spaced bins across a pre-determined range large enough to accommodate all potential data sets. Then upper gate values were determined for each bin as the 99th percentile of the DamFRET distribution in that bin. For bins at very low and very high acceptor intensities, there are not enough cells to accurately calculate this gate value. As a result, for bins above the 99th acceptor percentile and bins below 2 million acceptor intensity units, the upper gate value was set to the value of the nearest valid bin. The upper gate profile was then smoothed by boxcar smoothing with a width of 5 bins and shifted upwards by 0.028 DAmFRET units to ensure that the negative control signal lies completely within the negative FRET gate. The lower gate values for all bins were set equal to –0.2 DamFRET units. For all samples, then, cells falling above this negative FRET gate can be said to contain assembled (FRET-positive) protein. A metric reporting the gross percentage of the expressing cells containing assembled proteins is therefore reported as fgate which is a unitless statistic between 0 and 1.

This gate is then applied to all DAmFRET plots to define cells containing proteins that are either positive (self-assembled) or negative (monomeric). In each of the 64 gates, the fraction of cells in the assembled population were plotted as a ratio to total cells in the gate.

## Microscopy

Cells were imaged using a CSU-W1 spinning disc Ti2 microscope (Nikon) and visualized through a 100 x Plan Apochromat objective (NA 1.45). mEos3.1 was excited at 488 nm and emission was collected for 50ms per frame through a ET525/36 M bandpass filter on to a Flash 4 camera (Hamamatsu). Full z stacks of all cells were acquired over ~15 μm total distance with z spacing of 0.2 μm. Transmitted light was collected at the middle of the z stack for reference. To quantify the total intensity of each cell, the z stacks were processed using Fiji (https://imagej.net/software/fiji/). Images were first converted to 32-bit and sum projected in Z. Regions of interest (ROIs) were hand drawn around each cell using the ellipse tool in Fiji on the transmitted light image. These ROIs were then used to measure the area, mean, standard deviation, and integrated density of each cell on the fluorescence channel. Each cell was then classified as being diffuse or punctate by calculating the coefficient of variance (CV, standard deviation divided by the square root of the mean intensity) of the fluorescence. Cells were only considered punctate if their CV was greater than 30. Cells from each category that had equivalent integrated densities were directly compared and the images were plotted on the same intensity scale. The volume of each cell was calculated by fitting the transmitted light hand drawn ROI to an ellipse. The cell was then assumed to be a symmetric ellipsoid with the parameters of the fit ellipse from Fiji. The volume was calculated by 4/3*pi*major*minor*minor, where major and minor are the major and minor axes of the Fiji ellipse fit to the hand drawn ellipse ROI. The volumes reported are the volumes of the 3D ellipsoids. Concentrations were calculated by dividing the integrated densities by the calculated volume in μm$^3$, yielding units of fluorescence per μm$^3$.

## SDD-AGE

Semi-denaturing detergent agarose gel electrophoresis (SDD-AGE) was done as in *Khan et al., 2018*. The gel was imaged directly using a GE Typhoon Imaging System using a 488 laser and 525(40) BP filter. Images were then loaded into ImageJ for contrast adjustment. Images were gaussian blurred with a radius of 1 and then background subtracted with a 200 pixel rolling ball. Representative samples were then cropped from the original image for emphasis. Dotted or solid lines denote where different regions of the same gel were aligned for comparison. Line profiles of the protein smear were quantified using the following procedure. All images were processed in Fiji (https://imagej.net/software/fiji/). Gel images were first background subtracted using a rolling ball with a radius of 200 pixels. The images were then rotated so that the orientation was perfectly aligned. A user-defined line was drawn on the image down the first lane. A 10 pixel wide line profile was generated using an in house written plugin 'polyline kymograph jru v1'. This line was then programmatically shifted to every other lane and line profiles were generated for each. Each line profile was then normalized to the integral under the line profile using 'normalize trajectories jru v1'. From these line profiles, csv sheets were generated and these were imported to python to make line profile plots.

## Amyloid prediction

Comparative analyses were performed using default settings at the listed, public web servers as were available on April 1st, 2021. See *Supplementary file 2* for further details.

## Molecular Simulations

The simulations were carried out using the AMBER 20 package (*Case et al., 2020*) with ff14SB force field (*Maier et al., 2015*) and explicit TIP3P water model (*Price and Brooks, 2004*). The aggregates were placed in a cubic solvation box. In each aggregate, the minimum distance from the aggregate to the box boundary was set to be 12 nm to avoid self-interactions. An 8 Å cutoff was applied for the nonbonded interactions, and the particle mesh Ewald (PME) method (*Essmann et al., 1995*) was used to calculate the electrostatic interaction with cubic-spline interpolation and a grid spacing of approximately 1 Å. Once the box was set up, we performed a structural optimization with aggregates fixed, and on a second step allowed them to move, followed by a graduate heating procedure (NVT) with individual steps of 200 ps from 0 K to 300 K. Finally, the production runs were carried out using NPT Langevin dynamics with constant pressure of 1 atm at 300 K.

## Acknowledgements

We thank anonymous reviewers of the Review Commons platform for providing invaluable feedback prior to this manuscript's submission. We thank Viet Man and Jonathon Russell for assistance with molecular simulations; Patrick van der Wel, Ansgar Siemer, Rohit Pappu and Wei-feng Xue for helpful discussions; and Mark Miller, Alejandro Rodriguez Gama, and Megan Halfmann for assistance with figure preparation. We thank Alexander I Alexandrov for plasmids encoding $Q_4X$ repeats, early results from which stimulated this project's inception. This work was funded by the National Institute Of General Medical Sciences of the National Institutes of Health under Award Number R01GM130927 (to RH) and the Stowers Institute for Medical Research. A portion of this work was done to fulfill, in part, requirements for a PhD thesis research for TSK as a student registered with the Open University, UK at the Stowers Institute for Medical Research Graduate School, USA. Original data underlying this manuscript can be accessed from the Stowers Original Data Repository at http://www.stowers.org/research/publications/libpb-1494.

---

## Additional information

### Funding

| Funder | Grant reference number | Author |
| --- | --- | --- |
| National Institutes of Health | R01GM130927 | Randal Halfmann |

---

| Funder | Grant reference number | Author |
|---|---|---|

The funders had no role in study design, data collection and interpretation, or the decision to submit the work for publication.

## Author contributions

Tej Kandola, Data curation, Formal analysis, Investigation, Visualization, Methodology, Writing – original draft; Shriram Venkatesan, Data curation, Formal analysis, Investigation, Methodology, Writing - review and editing; Jiahui Zhang, Formal analysis, Investigation, Visualization; Brooklyn T Lerbakken, Alex Von Schulze, Jillian F Blanck, Jianzheng Wu, Investigation; Jay R Unruh, Formal analysis, Investigation, Methodology; Paula Berry, Jeffrey J Lange, Formal analysis, Investigation; Andrew C Box, Investigation, Methodology; Malcolm Cook, Methodology, Writing - review and editing; Celeste Sagui, Conceptualization, Formal analysis, Supervision, Visualization, Writing - review and editing; Randal Halfmann, Conceptualization, Supervision, Funding acquisition, Visualization, Methodology, Writing – original draft, Project administration, Writing - review and editing

## Author ORCIDs

Shriram Venkatesan http://orcid.org/0000-0003-0778-2474
Jay R Unruh http://orcid.org/0000-0003-3077-4990
Jeffrey J Lange http://orcid.org/0000-0003-4970-6269
Randal Halfmann http://orcid.org/0000-0002-6592-1471

Reviewer #1 (Public Review): https://doi.org/10.7554/eLife.86939.3.sa1
Reviewer #2 (Public Review): https://doi.org/10.7554/eLife.86939.3.sa2
Author Response https://doi.org/10.7554/eLife.86939.3.sa3

# Additional files

## Supplementary files

• Supplementary file 1. List of plasmids and sequences used in this study.

• Supplementary file 2. Summary of results from publically available amyloid propensity predictor web servers. All methods available as of April 2021.

• MDAR checklist

## Data availability

Original data underlying this manuscript can be accessed from the Stowers Original Data Repository at http://www.stowers.org/research/publications/libpb-1494.

The following dataset was generated:

| Author(s) | Year | Dataset title | Dataset URL | Database and Identifier |
|---|---|---|---|---|
| Kandola T, Venkatesan S, Lerbakken B, Blanck JF, Wu J, Unruh J, Box A, Cook M, Halfmann R | 2023 | Pathologic polyglutamine aggregation begins with a self-poisoning polymer crystal | http://www.stowers.org/research/publications/libpb-1494 | Stowers Original Data Repository, LIBPB-1494 |

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
