## [Editor Report · eLife assessment]

The authors investigate the mechanism of amyloid nucleation in a cellular system using novel ratiometric measurements, providing **fundamental** insight into the role of polyglutamine length and the sequence features of glutamine-rich regions in amyloid formation. The problem addressed by this study is very significant and the ability to assess nucleation in cells is of considerable value. The data, as presented and analyzed, are mostly **convincing**.

---

## [Referee Report · Reviewer #1 (Public Review)]

Numerous neurodegenerative diseases are thought to be driven by the aggregation of proteins into insoluble filaments known as "amyloids". Despite decades of research, the mechanism by which proteins convert from the soluble to insoluble state is poorly understood. In particular, the initial nucleation step is has proven especially elusive to both experiments and simulation. This is because the critical nucleus is thermodynamically unstable, and therefore, occurs too infrequently to directly observe. Furthermore, after nucleation much faster processes like growth and secondary nucleation dominate the kinetics, which makes it difficult to isolate the effects of the initial nucleation event. In this work Kandola et al. attempt to surmount these obstacles using individual yeast cells as microscopic reaction vessels. The large number of cells, and their small size, provides the statistics to separate the cells into pre- and post-nucleation populations, allowing them to obtain nucleation rates under physiological conditions. By systematically introducing mutations into the amyloid-forming polyglutamine core of huntingtin protein, they deduce the probable structure of the amyloid nucleus. This work shows that, despite the complexity of the cellular environment, the seemingly random effects of mutations can be understood with a relatively simple physical model. Furthermore, their model shows how amyloid nucleation and growth differ in significant ways, which provides testable hypotheses for probing how different steps in the aggregation pathway may lead to neurotoxicity.

In this study Kandola et al. probe the nucleation barrier by observing a bimodal distribution of cells that contain aggregates; the cells containing aggregates have had a stochastic fluctuation allowing the proteins to surmount the barrier, while those without aggregates have yet to have a fluctuation of suitable size. The authors confirm this interpretation with the selective manipulation of the PIN gene, which provides an amyloid template that allows the system to skip the nucleation event.

In simple systems lacking internal degrees of freedom (i.e., colloids or rigid molecules) the nucleation barrier comes from a significant entropic cost that comes from bringing molecules together. In large aggregates this entropic cost is balanced by attractive interactions between the particles, but small clusters are unable to form the extensive network of stabilizing contacts present in the larger aggregates. Therefore, the initial steps in nucleation incur an entropic cost without compensating attractive interactions (this imbalance can be described as a surface tension). When internal degrees of freedom are present, such as the conformational states of a polypeptide chain, there is an additional contribution to the barrier coming from the loss of conformational entropy required to the adopt aggregation-prone state(s). In such systems the clustering and conformational processes do not necessarily coincide, and a major challenge studying nucleation is to separate out these two contributions to the free energy barrier. Surprisingly, Kandola et al. find that the critical nucleus occurs within a single molecule. This means that the largest contribution to the barrier comes from the conformational entropy cost of adopting the beta-sheet state. Once this state is attained, additional molecules can be recruited with a much lower free energy barrier.

There are several considerations in interpreting this result. First, the height of the nucleation barrier(s) comes from the relative strength of the entropic costs compared to the binding affinities. This balance determines how large a nascent nucleus must grow before it can form interactions comparable to a mature aggregate. In amyloid nuclei the first three beta strands form immature contacts consisting of either side chain or backbone contacts, whereas the fourth strand is the first that is able to form both kinds of contacts (as in a mature fibril). This study used relatively long polypeptides of 60 amino acids. This is greater than the 20-40 amino acids found in amyloid-forming molecules like ABeta or IAPP. As a result, Kandola et al.'s molecules are able to fold enough times to create four beta strands and generate mature contacts intramolecularly. This authors make the plausible claim that these intramolecular folds explain the well-known length threshold (L~35) observed in polyQ diseases. The intramolecular folds reduce the importance of clustering multiple molecules together and increase the importance of the conformational states. Similarly, manipulating the sequence or molecular concentrations will be expected to manipulate the relative magnitude of the binding affinities and the clustering entropy, which will shift the relative heights of the entropic barriers.

The authors make an important point that the structure of the nucleus does not necessarily resemble that of the mature fibril. They find that the critical nucleus has a serpentine structure that is required by the need to form four beta strands to get the first mature contacts. However, this structure comes at a cost because residues in the hairpins cannot form strong backbone or zipper interactions. Mature fibrils offer a beta sheet template that allows incoming molecules to form mature contacts immediately. Thus, it is expected that the role of the serpentine nucleus is to template a more extended beta sheet structure that is found in mature fibrils.

A second point of consideration is the striking homogeneity of the nucleus structure they describe. This homogeneity is likely to be somewhat illusory. Homopolymers, like polyglutamine, have a discrete translational symmetry, which implies that the hairpins needed to form multiple beta sheets can occur at many places along the sequence. The asparagine residues introduced by the authors place limitations on where the hairpins can occur, and should be expected to increase structural homogeneity. Furthermore, the authors demonstrate that polyglutamine chains close to the minimum length of ~35 will have strict limitations on where the folds must occur in order to attain the required four beta strands.

A novel result of this work is the observation of multiple concentration regimes in the nucleation rate. Specifically, they report a plateau-like regime at intermediate regimes in which the nucleation rate is insensitive to protein concentration. The authors attribute this effect to the "self-poisoning" phenomenon observed in growth of some crystals. This is a valid comparison because the homogeneity observed in NMR and crystallography structures of mature fibrils resemble a one-dimensional crystal. Furthermore, the typical elongation rate of amyloid fibrils (on the order of one molecule per second) is many orders of magnitude slower than the molecular collision rate (by factors of 10^6 or more), implying that the search for the beta-sheet state is very slow. This slow conformational search implies the presence of deep kinetic traps that would be prone to poisoning phenomena. However, the observation of poisoning in nucleation during nucleation is striking, particularly in consideration of the expected disorder and concentration sensitivity of the nucleus. Kandola et al.'s structural model of an ordered, intramolecular nucleus explains why the internal states responsible for poisoning are relevant in nucleation.

To achieve these results the authors used a novel approach involving a systematic series of simple sequences. This is significant because, while individual experiments showed seemingly random behavior, the randomness resolved into clear trends with the systematic approach. These trends provided clues to build a model and guide further experiments.

There has been discussion in the review process about whether a monomeric nucleus is consistent with established properties of huntingtin aggregation. I do not see a problem with an energetically unfavorable conformational state preceding a concentration-dependent growth step. The authors make the case for this sequence using a schematic free energy landscape (Fig 6) that has many similarities to a free energy landscape derived from models of polyQ nucleation (Phan et al. 2022, see Fig. 6). The theory does not consider molecules large enough to form the conformational state described by Kandola et al., but the transition state is otherwise very similar.

---

## [Referee Report · Reviewer #2 (Public Review)]

Kandola et al. explore the important and difficult question regarding the initiating event that triggers (nucleates) amyloid fibril growth in glutamine-rich domains. The researchers use a fluorescence technique that they developed, dAMFRET, in a yeast system where they can manipulate the expression level over several orders of magnitude, and they can control the length of the polyglutamine domain as well as the insertion of interfering non-glutamine residues. Using flow cytometry, they can interrogate each of these yeast 'reactors' to test for self-assembly.

In the introduction, the authors provide a fairly thorough yet succinct review of the relevant literature into the mechanisms of polyglutamine-mediated aggregation over the last two decades, as well as a fairly clear description of the experimental techniques they developed.

Their assay shows that the fraction of cells with AmFRET signal increases strongly with an increase in polyQ length, with a threshold around 35-40 glutamines. This roughly correlates with the Q-length dependence of disease. The experiments in which asparagine or other amino acids are inserted at variable positions in the glutamine repeat are creative and thorough, and the data along with the simulations provide compelling support for the proposed Q zipper model. The experiments are strongly supportive of a model where formation of the beta-sheet nucleus is within a monomer. This is a potentially important result, as there are conflicting data in the literature as to whether the nucleus in polyQ is monomer.

The authors present convincing data that there are differences in the structural stability of their "QU" versus "QB" aggregates. However, the conclusion that "QB" must have multilamellar architecture versus "QU" was feasible but less compelling.

The authors present intriguing data showing that amyloid formation does not monotonically increase with increasing concentration, and their conclusion that high concentrations of polyQ can 'self-poison' amyloid growth is supported by the experimental data. The discussion surrounding the mechanism by which 'self-poisoning' occurs is confusing. The authors variously discuss that soluble oligomers must be the inhibitory species, that dead-end products of Q zipper nuclei are the inhibitory species, or that self-poisoning occurs because conformational conversion at the templating surface is slow relative to the rate of arrival of new molecules to the surface. The data seem consistent with an argument that, at high concentrations, non-structured polyQ oligomers form which interfere with elongation into structured amyloid assemblies - but it is not clear why such oligomers would be zippers.

Overall, this is a very valuable and thorough exploration of the fundamental question as to the nature and identity of the nucleating species in polyglutamine aggregation.

---

## [Author Response]

The following is the authors’ response to the current reviews.

We thank both reviewers for their detailed and positive assessment of our work.

To Reviewer #2, we have now explicated the pattern -- (QXQXQX>3)4 where X>3 denotes any length of three or more residues of any composition -- in the first paragraph of the discussion.

To Reviewer #3, we have made slight modifications to the text in the “Q zippers poison themselves” results section, to attempt to further clarify the mechanism of self-poisoning.

Briefly, the reviewer questions if an alternative model -- where inhibition involves non-structured rather than Q-zipper containing oligomers -- better explains the data. We provided two lines of evidence that we believe exclude this alternative model. First, we point out in the first paragraph of the “Q zippers poison themselves” section that the cells that unexpectedly lack amyloid in the high concentration regime have negligible levels of AmFRET, indicating that the inhibitory oligomers themselves occur at low concentrations regardless of the total concentration, and are therefore limited by a kinetic barrier. Second, we point out in the third paragraph of the section that the severity of amyloid inhibition with respect to concentration has a sequence dependence that matches the expectation of converging phase boundaries for crystal polymorphs -- specifically, inhibition is most severe for sequences that have a local Q density just high enough to form a Q zipper on both sides of each strand. Inhibition relaxed for sequences having more or less Qs than that threshold. In contrast, disordered oligomerization is not expected to have such a dependence on the precise pattern of Qs and Ns.

The following is the authors’ response to the original reviews.

We are pleased that the editors find our study valuable. We find that the reviewers’ criticisms largely arise from misunderstandings inherent to the conceptually challenging nature of the topic, rather than fundamental flaws, as we will elaborate here. We are grateful for the opportunity afforded by eLife to engage reviewers in what we intend to be a constructive public dialogue.

**Response to Reviewer 1**

This review is highly critical but lacks specifics. The reviewer’s criticisms reflect a position that seems to dismiss a critical role for (or perhaps even the existence of) conformational ordering in polyQ amyloid, which is untenable.

The reviewer states that our objective to characterize the amyloid nucleus “rests on the assertion that polyQ forms amyloid structures to the exclusion of all other forms of solids”. We do not fully agree with this assertion because our findings show that detectable aggregation is rate-limited by conformational ordering, as evident by (1) its discontinuous relationship to concentration, (2) its acceleration by a conformational template, and (3) its strict dependence on very specific sequence features that are consistent with amyloid structure but not disordered aggregation.

We strongly disagree with the reviewer’s subjective statement that we have not critically assessed our findings and that they do not stand up to scrutiny. This statement seems to rest on the perceived contradiction of our findings with that of Crick et al. 2013. Contrary to the reviewer’s assessment, we argue here that the conclusions of Crick et al. do more to support than to refute our findings. Briefly, Crick et al. investigated the aggregation of synthetic Q30 and Q40 peptides in vitro, wherein fibrils assembled from high concentrations of peptide were demonstrated to have saturating concentrations in the low micromolar range. As explained below, this finding of a saturating concentration does not refute our results. More relevant to the present work are their findings that “oligomers” accumulated over an hours-long timespan in solutions that are subsaturated with respect to fibrils, and these oligomers themselves have (nanomolar) critical concentrations. The authors postulated that the oligomers result from liquid–liquid demixing of intrinsically disordered polyglutamine. However, phase separation by a peptide is expected to fix its concentration in both the solute and condensed phases, and, because disordered phase separation is faster than amyloid formation, the postulated explanation removes the driving force for any amyloid phase with a critical solubility greater than that of the oligomers. In place of this interpretation that truly does appear to -- in the reviewer’s words -- “contradict basic physical principles of how homopolymers self-assemble”, we interpret these oligomers as evidence of Q zipper-containing self-poisoned multimers, rounded as an inherent consequence of self-poisoning (Ungar et al., 2005), and plausibly akin to semicrystalline spherulites that have been observed in other polymer crystal and amyloid-forming systems (Crist and Schultz, 2016; Vetri and Foderà, 2015). Importantly, the physical parameters governing the transition between amyloid spherulites and fibrils have been characterized in the case of insulin (Smith et al. 2012), where it was found that spherulites form at lower protein concentrations than fibrils. This mirrors the observation by Crick et al. that fibrils have a higher solubility limit than the spherical oligomers..Further rebuttal to the perceived incompatibility of monomeric nucleation with the existence of a critical concentration for amyloid

We appreciate that the concept of a monomeric nucleus can superficially appear inconsistent with the fact that crystalline solids such as polyQ amyloid have a saturating concentration, but this is only true if one neglects that polyQ amyloids are polymer crystals with intramolecular ordering. The perceived discrepancy is perhaps most easily dispelled by the fact that folded proteins can form crystals, and the folded state of the protein. These crystals have critical concentrations, and the protein subunits within them each have intramolecular crystalline order (in the form of secondary structure). When placed in a subsaturated solution, the protein crystals dissolve into the constituent monomers, and yet those monomers still retain intramolecular order. Our present findings for polyQ are conceptually no different.

To further extrapolate this simple example to polyQ, one can also draw on the now well-established phenomenon of secondary nucleation, whereby transient interactions of soluble species with ordered species leads to their own ordering (Törnquist et al., 2018). Transience is important here because it implies that intramolecular ordering can in principle propagate even in solutions that are subsaturated with respect to bulk crystallization. This is possible in the present case because the pairing of sufficiently short beta strands (equivalent to “stems” in the polymer crystal literature) will be more stable intramolecularly than intermolecularly, due to the reduced entropic penalty of the former. Our elucidation that Q zipper ordering can occur with shorter strands intramolecularly than intermolecularly (Fig. S4C-D) demonstrates this fact. It is also evident from published descriptions of single molecule “crystals” formed in sufficiently dilute solutions of sufficiently long polymers (Hong et al., 2015; Keller, 1957; Lauritzen and Hoffman, 1960).

In suggesting that a saturating concentration for amyloid rules out monomeric nucleation, the reviewer assumes that the Q zipper-containing monomer must be stable relative to the disordered ensemble. This is not inherent to our claim. The monomeric nucleating structure need not be more stable than the disordered state, and monomers may very well be disordered at equilibrium at low concentrations. To be clear, our claim requires that the Q zipper-containing monomer is both on pathway to amyloid and less stable than all subsequent species that are on pathway to amyloid. The former requirement is supported by our extensive mutational analysis. The latter requirement is supported by our atomistic simulations showing the Q zipper-containing monomer is stabilized by dimerization (included in our 2021 preprint). Hence, requisite ordering in the nucleating monomer is stabilized by intermolecular interactions. We provide in Author response image 1 an illustration to clarify what we believe to be the discrepancy between our claim and the reviewer’s interpretation.

That the rate-limiting fluctuation for a crystalline phase can occur in a monomer can also be understood as a consequence of Ostwald’s rule of stages, which describes the general tendency of supersaturated solutes, including amyloid forming proteins (Chakraborty et al., 2023), to populate metastable phases en route to more stable phases (De Yoreo, 2022; Schmelzer and Abyzov, 2017). Our findings with polyQ are consistent with a general mechanism for Ostwald’s rule wherein the relative stabilities of competing polymorphs differ with the number of subunits (De Yoreo, 2022; Navrotsky, 2004). As illustrated in Fig. 6 of Navrotsky, a polymorph that is relatively stable at small particle sizes tends to give way to a polymorph that -- while initially unstable -- becomes more stable as the particles grow. The former is analogous to our early stage Q zipper composed of two short sheets with an intramolecular interface, while the latter is analogous to the later stage Q zipper composed of longer sheets with an intermolecular interface. Subunit addition stabilizes the latter more than the former, hence the initial Q zipper that is stabilized more by intra- than intermolecular interactions will mature with growth to one that is stabilized more by intermolecular interactions.

We have added a new figure (Fig. 6) to the manuscript to illustrate qualitative features of the amyloid pathway we have deduced for polyQ.

Rebuttal to the perceived necessity of in vitro experiments

The overarching concern of this reviewer and reviewing editor is whether in-cell assays can inform on sequence-intrinsic properties. We understand this concern. We believe however that the relative merit of in-cell assays is largely a matter of perspective. The truly sequence-intrinsic behavior of polyQ, i.e. in a vacuum, is less informative than the “sequence-intrinsic” behaviors of interest that emerge in the presence of extraneous molecules from the appropriate biological context. In vitro experiments typically include a tiny number of these -- water, ions, and sometimes a crowding agent meant to approximate everything else. Obviously missing are the myriad quinary interactions with other proteins that collectively round out the physiological solvent. The question is what experimental context best approximates that of a living human neuron under which the pathological sequence-dependent properties of polyQ manifest. We submit that a living yeast cell comes closer to that ideal than does buffer in a test tube.

The reviewer’s statements that our findings must be validated in vitro ignores the fact -- stressed in our introduction -- that decades of in vitro work have not yet generated definitive evidence for or against any specific nucleus model. In addition to the above, one major problem concerns the large sizes of in vitro systems that obscure the effects of primary nucleation. For example, a typical in vitro experimental volume of e.g. 1.5 ml is over one billion-fold larger than the femtoliter volume of a cell. This means that any nucleation-limited kinetics of relevant amyloid formation are lost, and any alternative amyloid polymorphs that have a kinetic growth advantage -- even if they nucleate at only a fraction the rate of relevant amyloid -- will tend to dominate the system (Buell, 2017). Novel approaches are clearly needed to address these problems. We present such an approach, stretch it to the limit (as the reviewer notes) across multiple complementary experiments, and arrive at a novel finding that is fully and uniquely consistent with all of our own data as well as the collective prior literature.

That the preceding considerations are collectively essential to understand relevant amyloid behavior is evident from recent cryoEM studies showing that in vitro-generated amyloid structures generally differ from those in patients (Arseni et al., 2022; Bansal et al., 2021; Radamaker et al., 2021; Schmidt et al., 2019; Schweighauser et al., 2020; Yang et al., 2022). This is highly relevant to the present discourse because each amyloid structure is thought to emanate from a different nucleating structure. This means that in vitro experiments have broadly missed the mark in terms of the relevant thermodynamic parameters that govern disease onset and progression. Note that the rules laid out via our studies are not only consistent with structural features of polyQ amyloid in cells, but also (as described in the discussion) explain why the endogenous structure of a physiologically relevant Q zipper amyloid differs from that of polyQ.

A recent collaboration between the Morimoto and Knowles groups (Sinnige et al.) investigated the kinetics of aggregation by Q40-YFP expressed in *C. elegans* body wall muscle cells, using quantitative approaches that have been well established for in vitro amyloid-forming systems of the type favored by the reviewer. They calculate a reaction order of just 1.6, slightly higher than what would be expected for a monomeric nucleus but nevertheless fully consistent with our own conclusions when one accounts for the following two aspects of their approach. First, the polyQ tract in their construct is flanked by short poly-Histidine tracts on both sides. These charges very likely disfavor monomeric nucleation because all possible configurations of a four-stranded bundle position the beginning and end of the Q tract in close proximity, and Q40 is only just long enough to achieve monomeric nucleation in the absence of such destabilization. Second, the protein is fused to YFP, a weak homodimer (Landgraf et al., 2012; Snapp et al., 2003). With these two considerations, our model -- which was generated from polyQ tracts lacking flanking charges or an oligomeric fusion -- predicts that amyloid nucleation by their construct will occur more frequently as a dimer than a monomer. Indeed, their observed reaction order of 1.6 supports a predominantly dimeric nucleus. Like us and others, Sinnige et al. did not observe phase separation prior to amyloid formation. This is important because it not only argues against nucleation occurring in a condensate, it also suggests that the reaction order they calculated has not been limited by the concentration-buffering effect of phase separation.

While we agree that our conclusions rest heavily on DAmFRET data (for good reason), we do provide supporting evidence from molecular dynamics simulations, SDD-AGE, and microscopy.

To summarize, given the extreme limitations of in vitro experiments in this field, the breadth of our current study, and supporting findings from another lab using rigorous quantitative approaches, we feel that our claims are justified without in vitro data.

Rebuttals to other critiques

We do not deny that flanking domains can modulate the kinetics and stability of polyQ amyloid. However, as stated and referenced in the introduction, they do not appear to change the core structure. We have also added a paragraph concerning flanking domains to the discussion, and acknowledged that “the extent to which our findings will translate in these different contexts remains to be determined.” Nevertheless, that the intrinsic behavior of the polyQ tract itself is central to pathology is evident from the fact that the nine pathologic polyQ proteins have similar length thresholds despite different functions, flanking domains, interaction partners, and expression levels.

The reviewer states that we found nucleation potential to require 60 Qs in a row. Our data are collectively consistent with nucleation occurring at and above approximately 36 Qs, a point repeated in the paper. The reviewer may be referring to our statement, ”Sixty residues proved to be the optimum length to observe both the pre- and post-nucleated states of polyQ in single experiments”. The purpose of this statement is simply to describe the practical consideration that led us to use 60 Qs for the bulk of our assays. We do appreciate that the fraction of AmFRET-positive cells is very low for lengths just above the threshold, especially Q40. They are nevertheless highly significant (p = 0.004 in [PIN+] cells, one-tailed T-test), and we have modified the figure and text to clarify this.

The reviewer characterizes self-poisoning as the hallmark of crystallization from polymer melts, which would be problematic for our conclusions if self-poisoning were limited to this non-physiological context. In fact the term was first used to describe crystallization from solution (Organ et al., 1989), wherein the phenomenon is more pronounced (Ungar et al., 2005).

**Response to Reviewer 2**

We thank the reviewer for their detailed and helpful critique.

The reviewer correctly notes that the majority of our manipulations were conducted with 60-residue long tracts (which corresponds to disease onset in early adulthood), and this length facilitates intramolecular nucleation. However, we also analyzed a length series of polyQ spanning the pathological threshold, as well as a synthetic sequence designed explicitly to test the model nucleus structure with a tract shorter than the pathological threshold, and both experiments corroborate our findings.

The reviewer mentions “several caveats” that come with our result, but their subsequent elaboration suggests they are to be interpreted more as considerations than caveats. We agree that increasing sequence complexity will tend to increase homogeneity, but this is exactly the motivation of our approach. We explicitly set out to determine the minimal complexity sequence sufficient to specify the nucleating conformation, which we ultimately identified in terms of secondary and tertiary structure. We do not specify which parts of a long polyQ tract correspond to which parts of the structure, because, as the reviewer points out, they can occur at many places. Hence, depending on the length of the polyQ tract, the nucleus we describe may have any length of sequence connecting the strand elements. We do not think that the effects of N-residue placement can be interpreted as a confounding influence on hairpin position because the striking even-odd pattern we observe implicates the sides of beta strands rather than the lengths. Moreover, we observe this pattern regardless of the residue used (Gly, Ser, Ala, and His in addition to Asn).

We thank the reviewer for noting the novelty and plausibility of the self-poisoning connection. We would like to elaborate on our finding that self-poisoning inhibits nucleation (in addition to elongation), as this will be confusing to many readers. While self-poisoning is claimed to inhibit primary nucleation in the polymer crystal literature (Ungar et al., 2005; Zhang et al., 2018), the semantics of “nucleation” in this context warrants clarification. Technically, the same structure can be considered a nucleus in one context but not in another. The Q zipper monomer, even if it is rate-limiting for amyloid formation at low concentrations (and is therefore the “nucleus”), is not necessarily rate-limiting when self-poisoned at high concentrations. Whether it comprises the nucleus in this case depends on the rates of Q zipper formation relative to subunit addition to the poisoned state. If the latter happens slower than Q zipper formation de novo, it can be said that self-poisoning inhibits nucleation, regardless of whether the Q zipper formed. We suspect this to be the mechanism by which preemptive oligomerization blocks nucleation in the case of polyQ, though other mechanisms may be possible.

We believe the revised text also now incorporates the remaining suggestions of this reviewer, with two exceptions. (1) We retain the phrase “hidden pattern”, because we believe our data argue for a nucleus whose formation requires that Qs occur in a pattern that we now elaborate as (QXQXQX>3)4 where X>3 denotes any length of three or more residues of any composition. In amyloids formed from long polyQ molecules, the nucleus will involve any subset of 12 Qs that match this pattern. (2) We decided not to re-order the mansucript to discuss self-poisoning after establishing the monomer nucleus (even though we agree that doing so would improve the logical flow) because the interpretation of the data with respect to self-poisoning helps to establish critical strand lengths, and self-poisoning creates an anomaly in the DAmFRET data that is difficult to ignore. We add text clarifying that high local concentrations “effectively shifts the rate-limiting step to the growth of a higher order relatively-disordered species”.

**Response to Reviewer 3**

We thank the reviewer for their helpful comments.

We opted to retain Figures 1A and B because we think they are important for comprehending the subject and objectives of the study. We modified the former to attempt to make it more clear. We have also elaborated on DAmFRET as it is a relatively new approach that may be unfamiliar to many readers. Beyond this, we refer the reviewer and readers to our cited prior work describing the theory and interpretation of DAmFRET. Note that the y-axes of DAmFRET plots are not raw FRET but rather “AmFRET”, a ratio of FRET to total expression level. As explained thoroughly in our cited prior work, the discontinuity of AmFRET with expression level indicates that the high AmFRET-population formed via a disorder-to-order transition. When the query protein is predicted to be intrinsically disordered, the discontinuous transition to high AmFRET invariably (among hundreds of proteins tested in prior published and unpublished work) signifies amyloid formation as corroborated by SDD-AGE and tinctorial assays.

When performed using standard flow cytometry as in the present study, every AmFRET measurement corresponds to a cell-wide average, and hence does not directly inform on the distribution of the protein between different stoichiometric species. As there is only one fluorophore per protein molecule, monomeric nuclei have no signal. DAmFRET can distinguish cells expressing monomers from stable dimers from higher order oligomers (see e.g. Venkatesan et al. 2019), and we are therefore quite confident that AmFRET values of zero correspond to cells in which a vast majority of the respective protein is not in homo-oligomeric species (i.e. is monomeric or in hetero-complexes with endogenous proteins). The exact value of AmFRET, even for species with the same stoichiometry, will depend both on the effect of their respective geometries on the proximity of mEos3.1 fluorophores, and on the fraction of protein molecules in the species. Hence, we only attempt to interpret the plateau values of AmFRET (where the fraction of protein in an assembled state approaches unity) as directly informing on structure, as we did in Fig. S3A.

We believe that AmFRET decreases with longer polyQ because the mass fraction of fluorophore decreases in the aggregate, simply because the extra polypeptide takes up volume in the aggregate.

Yes, the fraction of positive cells in a discontinuous DAmFRET plot does increase with time. However, given the more laborious data collection and derivation of nucleation kinetics in a system with ongoing translation, especially across hundreds of experiments with other variables, ours is a snapshot measurement to approximately derive the relative contributions of intra- and intermolecular fluctuations to the nucleation barrier, rather than the barrier’s magnitude.

We have revised the tautological statement by removing “non-amyloid containing”.

Concerning the correlation of our data with the pathological length threshold -- as we state in the first results section, “Our data recapitulated the pathologic threshold -- Q lengths 35 and shorter lacked AmFRET, indicating a failure to aggregate or even appreciably oligomerize, while Q lengths 40 and longer did acquire AmFRET in a length and concentration-dependent manner”. Hence, most of our experiments were conducted with 60Q not because it resembles the pathological threshold, but rather because it was most convenient for DAmFRET experiments.

Self-poisoning is a widely observed and heavily studied phenomenon in polymer crystal physics, though it seems not yet to have entered the lexicon of amyloid biologists. We were new to this concept before it emerged as an extremely parsimonious explanation for our results. As described in the text, two pieces of evidence exclude the alternative mechanism suggested by the reviewer -- that non-structured oligomers form and subsequently engage and inhibit the template. Specifically, (1) inhibition occurs without any detectable FRET, even at high total protein concentration, indicating the species do not form in a concentration-dependent manner that would be expected of disordered oligomers; and (2) inhibition itself has strict sequence requirements that match those of Q zippers. Hence our data collectively suggest that inhibition is a consequence of the deposition of partially ordered molecules onto the templating surface.

We have softened the subheading and text of the relevant section in the discussion to more clearly indicate the speculative nature of our statements concerning the possible role of self-poisoned oligomers in toxicity.

We stand by our statement 'that kinetically arrested aggregates emerge from the same nucleating event responsible for amyloid formation', as this follows directly from self-poisoning.

Regarding the arguments for lateral and axial growth, we agree that the data are indirect. However, that polyQ forms lamellar amyloids both in vitro and in vivo is now established, so we do not feel it necessary to rigorously show that here. Nevertheless, we need to include this section primarily because it introduces the fact that ordering in polyQ amyloid occurs in the lateral as well as axial dimensions, and the onset of lateral ordering (lamellar growth) explains the very different behaviors of QU and QB sequences apparent on the DAmFRET plots. Ultimately, the two dimensions of growth are important to understand self-poisoning and maturation of the short nucleating zipper to amyloid.

References

Arseni D, Hasegawa M, Murzin AG, Kametani F, Arai M, Yoshida M, Ryskeldi-Falcon B. 2022. Structure of pathological TDP-43 filaments from ALS with FTLD. Nature 601:139–143. doi:10.1038/s41586-021-04199-3

Bansal A, Schmidt M, Rennegarbe M, Haupt C, Liberta F, Stecher S, Puscalau-Girtu I, Biedermann A, Fändrich M. 2021. AA amyloid fibrils from diseased tissue are structurally different from in vitro formed SAA fibrils. Nat Commun 12:1013. doi:10.1038/s41467-021-21129-z

Buell AK. 2017. The Nucleation of Protein Aggregates - From Crystals to Amyloid Fibrils. Int Rev Cell Mol Biol 329:187–226. doi:10.1016/bs.ircmb.2016.08.014

Chakraborty D, Straub JE, Thirumalai D. 2023. Energy landscapes of Aβ monomers are sculpted in accordance with Ostwald’s rule of stages. Sci Adv 9:eadd6921. doi:10.1126/sciadv.add6921Crist B, Schultz JM. 2016. Polymer spherulites: A critical review. Prog Polym Sci 56:1–63. doi:10.1016/j.progpolymsci.2015.11.006

De Yoreo JJ. 2022. Casting a bright light on Ostwald’s rule of stages. Proc Natl Acad Sci USA 119. doi:10.1073/pnas.2121661119

Hong Y, Yuan S, Li Z, Ke Y, Nozaki K, Miyoshi T. 2015. Three-Dimensional Conformation of Folded Polymers in Single Crystals. Phys Rev Lett 115:168301. doi:10.1103/PhysRevLett.115.168301Keller A. 1957. A note on single crystals in polymers: Evidence for a folded chain configuration. Philosophical Magazine 2:1171–1175. doi:10.1080/14786435708242746

Landgraf D, Okumus B, Chien P, Baker TA, Paulsson J. 2012. Segregation of molecules at cell division reveals native protein localization. Nat Methods 9:480–482. doi:10.1038/nmeth.1955

Lauritzen JI, Hoffman JD. 1960. Theory of Formation of Polymer Crystals with Folded Chains in Dilute Solution. J Res Natl Bur Stand A Phys Chem 64A:73–102. doi:10.6028/jres.064A.007

Navrotsky A. 2004. Energetic clues to pathways to biomineralization: precursors, clusters, and nanoparticles. Proc Natl Acad Sci USA 101:12096–12101. doi:10.1073/pnas.0404778101

Ohhashi Y, Ito K, Toyama BH, Weissman JS, Tanaka M. 2010. Differences in prion strain conformations result from non-native interactions in a nucleus. Nat Chem Biol 6:225–230. doi:10.1038/nchembio.306

Organ SJ, Ungar G, Keller A. 1989. Rate minimum in solution crystallization of long paraffins. Macromolecules 22:1995–2000. doi:10.1021/ma00194a078

Radamaker L, Baur J, Huhn S, Haupt C, Hegenbart U, Schönland S, Bansal A, Schmidt M, Fändrich M. 2021. Cryo-EM reveals structural breaks in a patient-derived amyloid fibril from systemic AL amyloidosis. Nat Commun 12:875. doi:10.1038/s41467-021-21126-2

Sahoo B, Singer D, Kodali R, Zuchner T, Wetzel R. 2014. Aggregation behavior of chemically synthesized, full-length huntingtin exon1. Biochemistry 53:3897–3907. doi:10.1021/bi500300c

Schmelzer JWP, Abyzov AS. 2017. How do crystals nucleate and grow: ostwald’s rule of stages and beyond In: Šesták J, Hubík P, Mareš JJ, editors. Thermal Physics and Thermal Analysis, Hot Topics in Thermal Analysis and Calorimetry. Cham: Springer International Publishing. pp. 195–211. doi:10.1007/978-3-319-45899-1_9

Schmidt M, Wiese S, Adak V, Engler J, Agarwal S, Fritz G, Westermark P, Zacharias M, Fändrich M. 2019. Cryo-EM structure of a transthyretin-derived amyloid fibril from a patient with hereditary ATTR amyloidosis. Nat Commun 10:5008. doi:10.1038/s41467-019-13038-z

Schweighauser M, Shi Y, Tarutani A, Kametani F, Murzin AG, Ghetti B, Matsubara T, Tomita T, Ando T, Hasegawa K, Murayama S, Yoshida M, Hasegawa M, Scheres SHW, Goedert M. 2020. Structures of α-synuclein filaments from multiple system atrophy. Nature 585:464–469. doi:10.1038/s41586-020-2317-6

Snapp EL, Hegde RS, Francolini M, Lombardo F, Colombo S, Pedrazzini E, Borgese N, Lippincott-Schwartz J. 2003. Formation of stacked ER cisternae by low affinity protein interactions. J Cell Biol 163:257–269. doi:10.1083/jcb.200306020

Törnquist M, Michaels TCT, Sanagavarapu K, Yang X, Meisl G, Cohen SIA, Knowles TPJ, Linse S. 2018. Secondary nucleation in amyloid formation. Chem Commun 54:8667–8684. doi:10.1039/c8cc02204f

Ungar G, Putra EGR, de Silva DSM, Shcherbina MA, Waddon AJ. 2005. The Effect of Self-Poisoning on Crystal Morphology and Growth Rates In: Allegra G, editor. Interphases and Mesophases in Polymer Crystallization I, Advances in Polymer Science. Berlin, Heidelberg: Springer Berlin Heidelberg. pp. 45–87. doi:10.1007/b107232

Vetri V, Foderà V. 2015. The route to protein aggregate superstructures: Particulates and amyloid-like spherulites. FEBS Lett 589:2448–2463. doi:10.1016/j.febslet.2015.07.006

Wild EJ, Boggio R, Langbehn D, Robertson N, Haider S, Miller JRC, Zetterberg H, Leavitt BR, Kuhn R, Tabrizi SJ, Macdonald D, Weiss A. 2015. Quantification of mutant huntingtin protein in cerebrospinal fluid from Huntington’s disease patients. The Journal of Clinical Investigation.

Yang Y, Arseni D, Zhang W, Huang M, Lövestam S, Schweighauser M, Kotecha A, Murzin AG, Peak-Chew SY, Macdonald J, Lavenir I, Garringer HJ, Gelpi E, Newell KL, Kovacs GG, Vidal R, Ghetti B, Ryskeldi-Falcon B, Scheres SHW, Goedert M. 2022. Cryo-EM structures of amyloid-β 42 filaments from human brains. Science 375:167–172. doi:10.1126/science.abm7285

Zhang X, Zhang W, Wagener KB, Boz E, Alamo RG. 2018. Effect of Self-Poisoning on Crystallization Kinetics of Dimorphic Precision Polyethylenes with Bromine. Macromolecules 51:1386–1397. doi:10.1021/acs.macromol.7b02745